# Interacting Contour Stochastic Gradient Langevin Dynamics

**Wei Deng[1,2], Siqi Liang[1], Botao Hao[3], Guang Lin[1], Faming Liang[1]**

[1]Purdue University [2]Morgan Stanley [3]DeepMind

fmliang@purdue.edu; weideng056@gmail.com

## Abstract

We propose an *interacting* contour stochastic gradient Langevin dynamics (IC-SGLD) sampler, an embarrassingly parallel multiple-chain contour stochastic gradient Langevin dynamics (CSGLD) sampler with *efficient interactions*. We show that ICSGLD can be *theoretically more efficient* than a single-chain CSGLD with an equivalent computational budget. We also present a novel random-field function, which facilitates the estimation of self-adapting parameters in big data and obtains free mode explorations. Empirically, we compare the proposed algorithm with popular benchmark methods for posterior sampling. The numerical results show a great potential of ICSGLD for large-scale uncertainty estimation tasks.

## 1 Introduction

Stochastic gradient Langevin dynamics (SGLD) (Welling & Teh, 2011) has achieved great successes in simulations of high-dimensional systems for big data problems. It, however, yields only a fast mixing rate when the energy landscape is simple, e.g., local energy wells are shallow and not well separated. To improve its convergence for the problems with complex energy landscapes, various strategies have been proposed, such as momentum augmentation (Chen et al., 2014; Ding et al., 2014), Hessian approximation (Ahn et al., 2012; Li et al., 2016), high-order numerical schemes (Chen et al., 2015; Li et al., 2019b), and cyclical learning rates (Izmailov et al., 2018; Maddox et al., 2019; Zhang et al., 2020b). In spite of their asymptotic properties in Bayesian inference (Vollmer et al., 2016) and non-convex optimization (Zhang et al., 2017), it is still difficult to achieve compelling empirical results for pathologically complex deep neural networks (DNNs).

To simulate from distributions with complex energy landscapes, e.g., those with a multitude of modes well separated by high energy barriers, an emerging trend is to run multiple chains, where interactions between different chains can potentially accelerate the convergence of the simulation. For example, Song et al. (2014) and Futami et al. (2020) showed theoretical advantages of appropriate interactions in ensemble/population simulations. Other multiple chain methods include particle-based nonlinear Markov (Vlasov) processes (Liu & Wang, 2016; Zhang et al., 2020a) and replica exchange methods (also known as parallel tempering) (Deng et al., 2021a). However, the particle-based methods result in an expensive kernel matrix computation given a large number of particles (Liu & Wang, 2016); similarly, naïvely extending replica exchange methods to population chains leads to a long waiting time to swap between non-neighboring chains (Syed et al., 2021). Therefore, how to conduct interactions between different chains, while maintaining the scalability of the algorithm, is the key to the success of the parallel stochastic gradient MCMC algorithms.

In this paper, we propose an interacting contour stochastic gradient Langevin dynamics (ICSGLD) sampler, a pleasingly parallel extension of contour stochastic gradient Langevin dynamics (CSGLD) (Deng et al., 2020b) with *efficient interactions*. The proposed algorithm requires minimal communication cost in that each chain shares with others the marginal energy likelihood estimate only. As a result, the interacting mechanism improves the convergence of the simulation, while the minimal communication mode between different chains enables the proposed algorithm to be naturally adapted to distributed computing with little overhead. For the single-chain CSGLD algorithm, despite its theoretical advantages as shown in Deng et al. (2020b), estimation of the marginal energy likelihood remains challenging for big data problems with a wide energy range, jeopardizing the empirical performance of the class of importance sampling methods (Gordon et al., 1993; Doucet et al., 2001;

Wang & Landau, 2001; Liang et al., 2007; Andrieu et al., 2010; Deng et al., 2020b) in big data applications. To resolve this issue, we resort to a novel interacting random-field function based on multiple chains for an ideal variance reduction and a more robust estimation. As such, we can greatly facilitate the estimation of the marginal energy likelihood so as to accelerate the simulations of notoriously complex distributions. To summarize, the algorithm has three main contributions:

- We propose a scalable interacting importance sampling method for big data problems with the minimal communication cost. A novel random-field function is derived to tackle the incompatibility issue of the class of importance sampling methods in big data problems.

- Theoretically, we study the local stability of a non-linear mean-field system and justify regularity properties of the solution of Poisson's equation. We also prove the asymptotic normality for the stochastic approximation process in mini-batch settings and show that ICSGLD is asymptotically more efficient than the single-chain CSGLD with an equivalent computational budget.

- Our proposed algorithm achieves appealing mode explorations using a fixed learning rate on the MNIST dataset and obtains remarkable performance in large-scale uncertainty estimation tasks.

## 2 PRELIMINARIES

### 2.1 STOCHASTIC GRADIENT LANGEVIN DYNAMICS

A standard sampling algorithm for big data problems is SGLD (Welling & Teh, 2011), which is a numerical scheme of a stochastic differential equation in mini-batch settings:

$$\mathbf{x}_{k+1} = \mathbf{x}_k - \epsilon_k \frac{N}{n} \nabla_{\mathbf{x}} \widetilde{U}(\mathbf{x}_k) + \sqrt{2\tau\epsilon_k} \boldsymbol{w}_k, \tag{1}$$

where $\mathbf{x}_k \in \mathcal{X} \in \mathbb{R}^d$, $\epsilon_k$ is the learning rate at iteration $k$, $N$ denotes the number of total data points, $\tau$ is the temperature, and $\boldsymbol{w}_k$ is a standard Gaussian vector of dimension $d$. In particular, $\frac{N}{n} \nabla_{\mathbf{x}} \widetilde{U}(\mathbf{x})$ is an unbiased stochastic gradient estimator based on a mini-batch data $\mathcal{B}$ of size $n$ and $\frac{N}{n} \widetilde{U}(\mathbf{x})$ is the unbiased energy estimator for the exact energy function $U(\mathbf{x})$. Under mild conditions on $U$, $\mathbf{x}_{k+1}$ is known to converge weakly to a unique invariant distribution $\pi(\mathbf{x}) \propto e^{-\frac{U(\mathbf{x})}{\tau}}$ as $\epsilon_k \to 0$.

### 2.2 CONTOUR STOCHASTIC GRADIENT LANGEVIN DYNAMICS

Despite its theoretical guarantees, SGLD can converge exponentially slow when $U(\mathbf{x})$ is non-convex and exhibits high energy barriers. To remedy this issue, CSGLD (Deng et al., 2020b) exploits the flat histogram idea and proposes to simulate from a flattened density with much lower energy barriers

$$\varpi_{\Psi_{\boldsymbol{\theta}}}(\mathbf{x}) \propto \pi(\mathbf{x})/\Psi_{\boldsymbol{\theta}}^{\zeta}(U(\mathbf{x})), \tag{2}$$

where $\zeta$ is a hyperparameter, $\Psi_{\boldsymbol{\theta}}(u) = \sum_{i=1}^{m} \left( \theta(i-1) e^{(\log\theta(i)-\log\theta(i-1))\frac{u-u_{i-1}}{\Delta u}} \right) 1_{u_{i-1} < u \le u_i}$.

In particular, $\{u_i\}_{i=0}^m$ determines the partition $\{\mathcal{X}_i\}_{i=1}^m$ of $\mathcal{X}$ such that $\mathcal{X}_i = \{\mathbf{x} : u_{i-1} < U(\mathbf{x}) \le u_i\}$, where $-\infty = u_0 < u_1 < \cdots < u_{m-1} < u_m = \infty$. For practical purposes, we assume $u_{i+1} - u_i = \Delta u$ for $i = 1, \cdots, m-2$. In addition, $\boldsymbol{\theta} = (\theta(1), \theta(2), \ldots, \theta(m))$ is the self-adapting parameter in the space $\boldsymbol{\Theta} = \left\{ (\theta(1), \cdots, \theta(m)) \, | \, 0 < \theta(1), \cdots, \theta(m) < 1 \& \sum_{i=1}^{m} \theta(i) = 1 \right\}$.

Ideally, setting $\zeta = 1$ and $\theta(i) = \theta_\infty(i)$, where $\theta_\infty(i) = \int_{\mathcal{X}_i} \pi(\mathbf{x}) d\mathbf{x}$ for $i \in \{1, 2, \cdots, m\}$, enables CSGLD to achieve a "random walk" in the space of energy and to penalize the over-visited partition (Wang & Landau, 2001; Liang et al., 2007; Fort et al., 2011; 2015). However, the optimal values of $\boldsymbol{\theta}_\infty$ is unknown *a priori*. To tackle this issue, CSGLD proposes the following procedure to adaptively estimate $\boldsymbol{\theta}$ via stochastic approximation (SA) (Robbins & Monro, 1951; Benveniste et al., 1990):

(1) Sample $\mathbf{x}_{k+1} = \mathbf{x}_k + \epsilon_k \frac{N}{n} \nabla_{\mathbf{x}} \widetilde{U}_{\Psi_{\boldsymbol{\theta}_k}}(\mathbf{x}_k) + \sqrt{2\tau\epsilon_k} \boldsymbol{w}_k$,

(2) Optimize $\boldsymbol{\theta}_{k+1} = \boldsymbol{\theta}_k + \omega_{k+1} \widetilde{\mathbb{H}}(\boldsymbol{\theta}_k, \mathbf{x}_{k+1})$,

**Algorithm 1** Interacting contour stochastic gradient Langevin dynamics algorithm (ICSGLD). $\{\mathcal{X}_i\}_{i=1}^m$ is pre-defined partition and $\zeta$ is a hyperparameter. The update rule in distributed-memory settings and discussions of hyperparameters is detailed in section B.1.1 in the supplementary material.

**[1.] (Data subsampling)** Draw a mini-batch data $\mathcal{B}_k$ from $\mathcal{D}$, and compute stochastic gradients $\nabla_{\mathbf{x}}\widetilde{U}(\mathbf{x}_k^{(p)})$ and energies $\widetilde{U}(\mathbf{x}_k^{(p)})$ for each $\mathbf{x}^{(p)}$, where $p \in \{1, 2, \cdots, P\}$, $|\mathcal{B}_k| = n$, and $|\mathcal{D}| = N$.

**[2.] (Parallel simulation)** Sample $\mathbf{x}_{k+1}^{\otimes P} := (\mathbf{x}_{k+1}^{(1)}, \mathbf{x}_{k+1}^{(2)}, \cdots, \mathbf{x}_{k+1}^{(P)})^{\top}$ based on SGLD and $\boldsymbol{\theta}_k$

$$\mathbf{x}_{k+1}^{\otimes P} = \mathbf{x}_k^{\otimes P} + \epsilon_k \frac{N}{n} \nabla_{\mathbf{x}} \widetilde{U}_{\Psi_{\boldsymbol{\theta}_k}}(\mathbf{x}_k^{\otimes P}) + \sqrt{2\tau\epsilon_k} \boldsymbol{w}_k^{\otimes P}, \tag{4}$$

where $\epsilon_k$ is the learning rate, $\tau$ is the temperature, $\boldsymbol{w}_k^{\otimes P}$ denotes $P$ independent standard Gaussian vectors, $\nabla_{\mathbf{x}}\widetilde{U}_{\Psi_{\boldsymbol{\theta}}}(\mathbf{x}^{\otimes P}) = (\nabla_{\mathbf{x}}\widetilde{U}_{\Psi_{\boldsymbol{\theta}}}(\mathbf{x}^{(1)}), \nabla_{\mathbf{x}}\widetilde{U}_{\Psi_{\boldsymbol{\theta}}}(\mathbf{x}^{(2)}), \cdots, \nabla_{\mathbf{x}}\widetilde{U}_{\Psi_{\boldsymbol{\theta}}}(\mathbf{x}^{(P)}))^{\top}$, and $\nabla_{\mathbf{x}}\widetilde{U}_{\Psi_{\boldsymbol{\theta}}}(\mathbf{x}) = \left[1 + \frac{\zeta\tau}{\Delta u}\left(\log\theta(J_{\widetilde{U}}(\mathbf{x})) - \log\theta((J_{\widetilde{U}}(\mathbf{x}) - 1) \vee 1)\right)\right]\nabla_{\mathbf{x}}\widetilde{U}(\mathbf{x})$ for any $\mathbf{x} \in \mathcal{X}$.

**[3.] (Stochastic approximation)** Update the self-adapting parameter $\theta(i)$ for $i \in \{1, 2, \cdots, m\}$

$$\theta_{k+1}(i) = \theta_k(i) + \omega_{k+1}\frac{1}{P}\sum_{p=1}^{P}\theta_k(J_{\widetilde{U}}(\mathbf{x}_{k+1}^{(p)}))\left(1_{i=J_{\widetilde{U}}(\mathbf{x}_{k+1}^{(p)})} - \theta_k(i)\right), \tag{5}$$

where $1_A$ is an indicator function that takes value 1 if the event $A$ appears and equals 0 otherwise.

where $\nabla_{\mathbf{x}}\widetilde{U}_{\Psi_{\boldsymbol{\theta}}}(\cdot)$ is a stochastic gradient function of $\varpi_{\Psi_{\boldsymbol{\theta}}}(\cdot)$ to be detailed in Algorithm 1. $\widetilde{\mathbb{H}}(\boldsymbol{\theta}, \mathbf{x}) := \left(\widetilde{\mathbb{H}}_1(\boldsymbol{\theta}, \mathbf{x}), \cdots, \widetilde{\mathbb{H}}_m(\boldsymbol{\theta}, \mathbf{x})\right)$ is random-field function where each entry follows

$$\widetilde{\mathbb{H}}_i(\boldsymbol{\theta}, \mathbf{x}) = \theta^{\zeta}(J_{\widetilde{U}}(\mathbf{x}))\left(1_{i=J_{\widetilde{U}}(\mathbf{x})} - \theta(i)\right), \text{ where } J_{\widetilde{U}}(\mathbf{x}) = \sum_{i=1}^{m} i 1_{u_{i-1} < \frac{N}{n}\widetilde{U}(\mathbf{x}) \le u_i}. \tag{3}$$

Theoretically, CSGLD converges to a sampling-optimization equilibrium in the sense that $\boldsymbol{\theta}_k$ approaches to a fixed point $\boldsymbol{\theta}_{\infty}$ and the samples are drawn from the flattened density $\varpi_{\Psi_{\boldsymbol{\theta}_{\infty}}}(\mathbf{x})$. Notably, the mean-field system is *globally stable* with a unique stable equilibrium point in a small neighborhood of $\boldsymbol{\theta}_{\infty}$. Moreover, such an appealing property holds even when $U(\mathbf{x})$ is non-convex.

## 3   INTERACTING CONTOUR STOCHASTIC GRADIENT LANGEVIN DYNAMICS

The major goal of interacting CSGLD (ICSGLD) is to improve the efficiency of CSGLD. In particular, the self-adapting parameter $\boldsymbol{\theta}$ is crucial for ensuring the sampler to escape from the local traps and traverse the whole energy landscape, and how to reduce the variability of $\boldsymbol{\theta}_k$'s is the key to the success of such a dynamic importance sampling algorithm. To this end, we propose an efficient variance reduction scheme via interacting parallel systems to improve the accuracy of $\boldsymbol{\theta}_k$'s.

### 3.1   INTERACTIONS IN PARALLELISM

Now we first consider a naïve parallel sampling scheme with $P$ chains as follows

$$\mathbf{x}_{k+1}^{\otimes P} = \mathbf{x}_k^{\otimes P} + \epsilon_k \frac{N}{n} \nabla_{\mathbf{x}} \widetilde{U}_{\Psi_{\boldsymbol{\theta}_k}}(\mathbf{x}_k^{\otimes P}) + \sqrt{2\tau\epsilon_k} \boldsymbol{w}_k^{\otimes P},$$

where $\mathbf{x}^{\otimes P} = (\mathbf{x}^{(1)}, \mathbf{x}^{(2)}, \cdots, \mathbf{x}^{(P)})^{\top}$, $\boldsymbol{w}_k^{\otimes P}$ denotes $P$ independent standard Gaussian vectors, and $\widetilde{U}_{\Psi_{\boldsymbol{\theta}}}(\mathbf{x}^{\otimes P}) = (\widetilde{U}_{\Psi_{\boldsymbol{\theta}}}(\mathbf{x}^{(1)}), \widetilde{U}_{\Psi_{\boldsymbol{\theta}}}(\mathbf{x}^{(2)}), \cdots, \widetilde{U}_{\Psi_{\boldsymbol{\theta}}}(\mathbf{x}^{(P)}))^{\top}$.

Stochastic approximation aims to find the solution $\boldsymbol{\theta}$ of the mean-field system $h(\boldsymbol{\theta})$ such that

$$h(\boldsymbol{\theta}) = \int_{\mathcal{X}} \widetilde{H}(\boldsymbol{\theta}, \mathbf{x}) \varpi_{\boldsymbol{\theta}}(d\mathbf{x}) = 0,$$

where $\varpi_{\boldsymbol{\theta}}$ is the invariant measure simulated via SGLD that approximates $\varpi_{\Psi_{\boldsymbol{\theta}}}$ in (2) and $\widetilde{H}(\boldsymbol{\theta}, \mathbf{x})$ is the novel random-field function to be defined later in (8). Since $h(\boldsymbol{\theta})$ is observable only up to large random perturbations (in the form of $\widetilde{H}(\boldsymbol{\theta}, \mathbf{x})$), the optimization of $\boldsymbol{\theta}$ based on isolated

random-field functions may not be efficient enough. However, due to the *conditional independence* of $\mathbf{x}^{(1)}, \mathbf{x}^{(2)}, \cdots, \mathbf{x}^{(P)}$ in parallel sampling, it is very natural to consider a Monte Carlo average

$$h(\boldsymbol{\theta}) = \frac{1}{P} \sum_{p=1}^{P} \int_{\mathcal{X}} \widetilde{H}(\boldsymbol{\theta}, \mathbf{x}^{(p)}) \varpi_{\boldsymbol{\theta}}(d\mathbf{x}^{(p)}) = 0. \tag{6}$$

Namely, we are considering the following stochastic approximation scheme

$$\boldsymbol{\theta}_{k+1} = \boldsymbol{\theta}_k + \omega_{k+1} \widetilde{\boldsymbol{H}}(\boldsymbol{\theta}_k, \mathbf{x}_{k+1}^{\otimes P}), \tag{7}$$

where $\widetilde{\boldsymbol{H}}(\boldsymbol{\theta}_k, \mathbf{x}_{k+1}^{\otimes P})$ is an interacting random-field function $\widetilde{\boldsymbol{H}}(\boldsymbol{\theta}_k, \mathbf{x}_{k+1}^{\otimes P}) = \frac{1}{P} \sum_{p=1}^{P} \widetilde{H}(\boldsymbol{\theta}_k, \mathbf{x}_{k+1}^{(p)})$. Note that the Monte Carlo average is very effective to reduce the variance of the interacting random-field function $\widetilde{\boldsymbol{H}}(\boldsymbol{\theta}, \mathbf{x}^{\otimes P})$ based on the conditionally independent random field functions. Moreover, each chain shares with others only a very short message during each iteration. Therefore, the interacting parallel system is well suited for distributed computing, where the *implementations and communication costs* are further detailed in section B.1.2 in the supplementary material. By contrast, each chain of the non-interacting parallel CSGLD algorithm deals with the parameter $\boldsymbol{\theta}$ and a large-variance random-field function $\widetilde{H}(\boldsymbol{\theta}, \mathbf{x})$ individually, leading to coarse estimates in the end.

Formally, for the population/ensemble interaction scheme (7), we define a novel random-field function $\widetilde{H}(\boldsymbol{\theta}, \mathbf{x}) = (\widetilde{H}_1(\boldsymbol{\theta}, \mathbf{x}), \widetilde{H}_2(\boldsymbol{\theta}, \mathbf{x}), \cdots, \widetilde{H}_m(\boldsymbol{\theta}, \mathbf{x}))$, where each component satisfies

$$\widetilde{H}_i(\boldsymbol{\theta}, \mathbf{x}) = \theta(J_{\widetilde{U}}(\mathbf{x})) \left( 1_{i=J_{\widetilde{U}}(\mathbf{x})} - \theta(i) \right). \tag{8}$$

As shown in Lemma 1, the corresponding mean-field function proposes to converge to a different fixed point $\boldsymbol{\theta}_\star$, s.t.

$$\theta_\star(i) \propto \left( \int_{\mathcal{X}_i} e^{-\frac{U(\mathbf{x})}{\tau}} d\mathbf{x} \right)^{\frac{1}{\zeta}} \propto \boldsymbol{\theta}_\infty^{\frac{1}{\zeta}}(i). \tag{9}$$

A large data set often renders the task of estimating $\boldsymbol{\theta}_\infty$ numerically challenging. By contrast, we resort to a different solution by estimating $\boldsymbol{\theta}_\star$ instead based on a large value of $\zeta$. The proposed algorithm is summarized in Algorithm 1. For more study on the scalablity of the new scheme, we leave the discussion in section B.1.3.

## 3.2 RELATED WORKS

Replica exchange SGLD (Deng et al., 2020a; 2021a) has successfully extended the traditional replica exchange (Swendsen & Wang, 1986; Geyer, 1991; Earl & Deem, 2005) to big data problems. However, it works with two chains only and has a low swapping rate. As shown in Figure 1(a), a naïve extension of multi-chain replica exchange SGLD yields low communication efficiency. Despite some recipe in the literature (Katzgraber et al., 2008; Bittner et al., 2008; Syed et al., 2021), how to conduct multi-chain replica exchange with low-frequency swaps is still an open question.

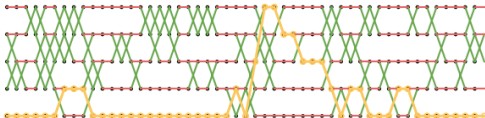
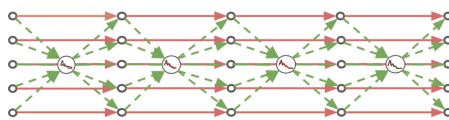

(a) Replica Exchange (parallel tempering)          (b) Interacting contour SGLD (ICSGLD)

Figure 1: A comparison of communication costs between replica exchange (RE) and ICSGLD. We see RE takes many iterations to swap with all the other chains; by contrast, ICSGLD possesses a pleasingly parallel mechanism where the only cost comes from sharing a light message.

Stein variational gradient descent (SVGD) (Liu & Wang, 2016) is a popular approximate inference method to drive a set of particles for posterior approximation. In particular, repulsive forces are proposed to prevent particles to collapse together into neighboring regions, which resembles our strategy of penalizing over-visited partition. However, SVGD tends to underestimate the uncertainty given a limited number of particles. Moreover, the quadratic cost in kernel matrix computation further raises the scalability concerns as more particles are proposed.

Admittedly, ICSGLD is not the first interacting importance sampling algorithm. For example, a population stochastic approximation Monte Carlo (pop-SAMC) algorithm has been proposed in Song et al. (2014), and an interacting particle Markov chain Monte Carlo (IPMCMC) algorithm has been proposed in Rainforth et al. (2016). A key difference between our algorithm and others is that our algorithm is mainly devised for big data problems. The IPMCMC and pop-SAMC are gradient-free samplers, which are hard to be adapted to high-dimensional big data problems.

Other parallel SGLD methods (Ahn et al., 2014; Chen et al., 2016) aim to reduce the computational cost of gradient estimations in distributed computing, which, however, does not consider interactions for accelerating the convergence. Li et al. (2019a) proposed asynchronous protocols to reduce communication costs when the master aggregates model parameters from all workers. Instead, we don't communicate the parameter $\mathbf{x} \in \mathbb{R}^d$ but only share $\boldsymbol{\theta} \in \mathbb{R}^m$ and the indices, where $m \ll d$.

Our work also highly resembles the well-known Federated Averaging (FedAvg) algorithm (Li et al., 2020; Deng et al., 2021b), except that the stochastic gradient $\widetilde{U}(\mathbf{x})$ is replaced with the random field function $\widetilde{H}(\boldsymbol{\theta}, \mathbf{x})$ and we only share the low-dimensional latent vector $\boldsymbol{\theta}$. Since privacy concerns and communication cost are not major bottlenecks of our problem, we leave the study of taking the Monte Carlo average in Eq.(6) every $K > 1$ iterations for future works.

## 4 CONVERGENCE PROPERTIES

To study theoretical properties of ICSGLD, we first show a local stability property that is well-suited to big data problems, and then we present the asymptotic normality for the stochastic approximation process in *mini-batch settings*, which eventually yields the desired result that ICSGLD is asymptotically more efficient than a single-chain CSGLD with an equivalent computational cost.

### 4.1 LOCAL STABILITY FOR NON-LINEAR MEAN-FIELD SYSTEMS IN BIG DATA

The first obstacle for the theoretical study is to approximate the components of $\boldsymbol{\theta}_\infty$ corresponding to the high energy region. To get around this issue, the random field function $\widetilde{H}(\boldsymbol{\theta}, \mathbf{x})$ in (8) is adopted to estimate a different target $\boldsymbol{\theta}_\star \propto \boldsymbol{\theta}_\infty^{\frac{1}{\zeta}}$. As detailed in Lemma 3 in the supplementary material, the mean-field equation is now formulated as follows

$$h_i(\boldsymbol{\theta}) \propto \theta_\star^\zeta(i) - (\theta(i)C_{\boldsymbol{\theta}})^\zeta + \text{perturbations}, \tag{10}$$

where $C_{\boldsymbol{\theta}} = \left( \frac{\widetilde{Z}_{\zeta,\boldsymbol{\theta}}}{\widetilde{Z}_{\zeta,\boldsymbol{\theta}_\star}^\zeta} \right)^{\frac{1}{\zeta}}$ and $\widetilde{Z}_{\zeta,\boldsymbol{\theta}} = \sum_{k=1}^m \frac{\int_{\mathcal{X}_k} \pi(\mathbf{x}) d\mathbf{x}}{\theta^{\zeta-1}(k)}$. We see that (10) may not be linearly stable as in Deng et al. (2020b). Although the solution of the mean-field system $h(\boldsymbol{\theta}) = 0$ is still unique, there may exist unstable invariant subspaces, leading us to consider the local properties. For a proper initialization of $\boldsymbol{\theta}$, which can be achieved by pre-training the model long enough time through SGLD, the mean value theorem implies a linear property in a local region

$$h_i(\boldsymbol{\theta}) \propto \theta_\star(i) - \theta(i) + \text{perturbations}.$$

Combining the perturbation theory (Vanden-Eijnden, 2001), we present the following stability result:

**Lemma 1 (Local stability, informal version of Lemma 3)** *Assume Assumptions A1-A4 (given in the supplementary material) hold. For any properly initialized $\boldsymbol{\theta}$, we have $\langle h(\boldsymbol{\theta}), \boldsymbol{\theta} - \widehat{\boldsymbol{\theta}}_\star \rangle \leq -\phi \|\boldsymbol{\theta} - \widehat{\boldsymbol{\theta}}_\star\|^2$, where $\widehat{\boldsymbol{\theta}}_\star = \boldsymbol{\theta}_\star + \mathcal{O}\left( \sup_\mathbf{x} \text{Var}(\xi_n(\mathbf{x})) + \epsilon + \frac{1}{m} \right)$, $\boldsymbol{\theta}_\star \propto \boldsymbol{\theta}_\infty^{\frac{1}{\zeta}}$, $\phi > 0$, $\epsilon$ denotes a learning rate, and $\xi_n(\mathbf{x})$ denotes the noise in the stochastic energy estimator of batch size $n$ and $\text{Var}(\cdot)$ denotes the variance.*

By justifying the drift conditions of the adaptive transition kernel and relevant smoothness properties, we can prove the existence and regularity properties of the solution of the Poisson's equation in Lemma 6 in the supplementary material. In what follows, we can control the fluctuations in stochastic approximation and eventually yields the $L^2$ convergence.

**Lemma 2 ($L^2$ convergence rate, informal version of Lemma 7)** *Given standard Assumptions A1-A5. $\boldsymbol{\theta}_k$ converges to $\widehat{\boldsymbol{\theta}}_\star$, where $\widehat{\boldsymbol{\theta}}_\star = \boldsymbol{\theta}_\star + \mathcal{O}\left( \sup_\mathbf{x} \text{Var}(\xi_n(\mathbf{x})) + \epsilon + \frac{1}{m} \right)$, such that*

$$\mathbb{E}\left[ \|\boldsymbol{\theta}_k - \widehat{\boldsymbol{\theta}}_\star\|^2 \right] = \mathcal{O}\left( \omega_k \right).$$

The result differs from Theorem 1 of Deng et al. (2020b) in that the biased fixed point $\widehat{\boldsymbol{\theta}}_\star$ instead of $\boldsymbol{\theta}_\star$ is treated as the equilibrium of the continuous system, which provides us a user-friendly proof. Similar techniques have been adopted by Durmus & Éric Moulines (2017); Xu et al. (2018). Although the global stability (Deng et al., 2020b) may be sacrificed when $\zeta \neq 1$ based on Eq.(8), $\boldsymbol{\theta}_\star \propto \boldsymbol{\theta}_\infty^{\frac{1}{\zeta}}$ is much easier to estimate numerically for any $i$ that yields $0 < \boldsymbol{\theta}_\infty(i) \ll 1$ based on a large $\zeta > 1$.

## 4.2 ASYMPTOTIC NORMALITY

To study the asymptotic behavior of $\omega_k^{-\frac{1}{2}}(\boldsymbol{\theta}_k - \widehat{\boldsymbol{\theta}}_\star)$, where $\widehat{\boldsymbol{\theta}}_\star$ is the equilibrium point s.t. $\widehat{\boldsymbol{\theta}}_\star = \boldsymbol{\theta}_\star + \mathcal{O}\left(\mathrm{Var}(\xi_n(\mathbf{x})) + \epsilon + \frac{1}{m}\right)$, we consider a fixed step size $\omega$ in the SA step for ease of explanation. Let $\bar{\boldsymbol{\theta}}_t$ denote the solution of the mean-field system in continuous time ($\bar{\boldsymbol{\theta}}_0 = \boldsymbol{\theta}_0$), and rewrite the single-chain SA step (7) as follows

$$
\begin{aligned}
\boldsymbol{\theta}_{k+1} - \bar{\boldsymbol{\theta}}_{(k+1)\omega} = {} & \boldsymbol{\theta}_k - \bar{\boldsymbol{\theta}}_{k\omega} + \omega \left( H(\boldsymbol{\theta}_k, \mathbf{x}_{k+1}) - H(\bar{\boldsymbol{\theta}}_{k\omega}, \mathbf{x}_{k+1}) \right) \\
& + \omega \left( H(\bar{\boldsymbol{\theta}}_{k\omega}, \mathbf{x}_{k+1}) - h(\bar{\boldsymbol{\theta}}_{k\omega}) \right) - \left( \bar{\boldsymbol{\theta}}_{(k+1)\omega} - \bar{\boldsymbol{\theta}}_{k\omega} - \omega h(\bar{\boldsymbol{\theta}}_{k\omega}) \right).
\end{aligned}
$$

Further, we set $\widetilde{\boldsymbol{\theta}}_{k\omega} := \omega^{-\frac{1}{2}}(\boldsymbol{\theta}_k - \bar{\boldsymbol{\theta}}_{k\omega})$. Then the stochastic approximation differs from the mean field system in that

$$
\widetilde{\boldsymbol{\theta}}_{(k+1)\omega} = \omega^{\frac{1}{2}} \underbrace{\sum_{i=0}^{k} \left( H(\boldsymbol{\theta}_i, \mathbf{x}_{i+1}) - H(\bar{\boldsymbol{\theta}}_{i\omega}, \mathbf{x}_{i+1}) \right)}_{\text{I: perturbations}} + \omega^{\frac{1}{2}} \sum_{i=0}^{k} \underbrace{\left( H(\bar{\boldsymbol{\theta}}_{i\omega}, \mathbf{x}_{i+1}) - h(\bar{\boldsymbol{\theta}}_{i\omega}) \right)}_{\text{II: martingale } \mathcal{M}_i} - \omega^{\frac{1}{2}} \cdot \text{remainder}
$$

$$
\approx \omega^{\frac{1}{2}} \sum_{i=0}^{k} h_{\boldsymbol{\theta}}(\boldsymbol{\theta}_{i\omega}) \underbrace{(\boldsymbol{\theta}_i - \bar{\boldsymbol{\theta}}_{i\omega})}_{\approx \omega^{\frac{1}{2}} \widetilde{\boldsymbol{\theta}}_{i\omega}} + \omega^{\frac{1}{2}} \sum_{i=0}^{k} \mathcal{M}_i \approx \int_0^{(k+1)\omega} h_{\boldsymbol{\theta}}(\bar{\boldsymbol{\theta}}_s) \widetilde{\boldsymbol{\theta}}_s ds + \int_0^{(k+1)\omega} \boldsymbol{R}^{\frac{1}{2}}(\bar{\boldsymbol{\theta}}_s) d\boldsymbol{W}_s,
$$

where $h_{\boldsymbol{\theta}}(\boldsymbol{\theta}) := \frac{d}{d\boldsymbol{\theta}} h(\boldsymbol{\theta})$ is a matrix, $\boldsymbol{W} \in \mathbb{R}^m$ is a standard Brownian motion, the last term follows from a certain central limit theorem (Benveniste et al., 1990) and $\boldsymbol{R}$ denotes the covariance matrix of the random-field function s.t. $\boldsymbol{R}(\boldsymbol{\theta}) := \sum_{k=-\infty}^{\infty} \mathrm{Cov}_{\boldsymbol{\theta}}(H(\boldsymbol{\theta}, \mathbf{x}_k), H(\boldsymbol{\theta}, \mathbf{x}_0))$.

We expect the weak convergence of $\boldsymbol{U}_k$ to the stationary distribution of a diffusion

$$
d\boldsymbol{U}_t = h_{\boldsymbol{\theta}}(\boldsymbol{\theta}_t)\boldsymbol{U}_t dt + \boldsymbol{R}^{1/2}(\boldsymbol{\theta}_t)d\boldsymbol{W}_t, \tag{11}
$$

where $\boldsymbol{U}_t = \omega_t^{-1/2}(\boldsymbol{\theta}_t - \widehat{\boldsymbol{\theta}}_\star)$. Given that $\boldsymbol{\theta}_t$ converges to $\widehat{\boldsymbol{\theta}}_\star$ sufficiently fast and the local linearity of $h_{\boldsymbol{\theta}}$, the diffusion (11) resembles the Ornstein–Uhlenbeck process and yields the following solution

$$
\boldsymbol{U}_t \approx e^{-t h_{\boldsymbol{\theta}}(\widehat{\boldsymbol{\theta}}_\star)} \boldsymbol{U}_0 + \int_0^t e^{-(t-s)h_{\boldsymbol{\theta}}(\widehat{\boldsymbol{\theta}}_\star)} \circ \boldsymbol{R}(\widehat{\boldsymbol{\theta}}_\star) d\boldsymbol{W}_s.
$$

Then we have the following theorem, whose formal proof is given in section C.3.

**Theorem 1 (Asymptotic Normality)** *Assume Assumptions A1-A5 (given in the supplementary material) hold. We have the following weak convergence*

$$
\omega_k^{-1/2}(\boldsymbol{\theta}_k - \widehat{\boldsymbol{\theta}}_\star) \Rightarrow \mathcal{N}(0, \boldsymbol{\Sigma}), \text{ where } \boldsymbol{\Sigma} = \int_0^{\infty} e^{th_{\boldsymbol{\theta}_\star}} \circ \boldsymbol{R} \circ e^{th_{\boldsymbol{\theta}_\star}^\top} dt, h_{\boldsymbol{\theta}_\star} = h_{\boldsymbol{\theta}}(\widehat{\boldsymbol{\theta}}_\star).
$$

## 4.3 INTERACTING PARALLEL CHAINS ARE MORE EFFICIENT

For clarity, we first denote an estimate of $\boldsymbol{\theta}$ based on ICSGLD with $P$ interacting parallel chains by $\boldsymbol{\theta}_k^P$ and denote the estimate based on a single-long-chain CSGLD by $\boldsymbol{\theta}_{kP}$.

Note that Theorem 1 holds for any step size $\omega_k = \mathcal{O}(k^{-\alpha})$, where $\alpha \in (0.5, 1]$. If we simply run a single-chain CSGLD algorithm with $P$ times of iterations, by Theorem 1,

$$
\omega_{kP}^{-1/2}(\boldsymbol{\theta}_{kP} - \widehat{\boldsymbol{\theta}}_\star) \Rightarrow \mathcal{N}(0, \boldsymbol{\Sigma}).
$$

As to ICSGLD, since the covariance $\boldsymbol{\Sigma}$ relies on $\boldsymbol{R}$, which depends on the covariance of the martingale $\{\mathcal{M}_i\}_{i \geq 1}$, the conditional independence of $\mathbf{x}^{(1)}, \mathbf{x}^{(2)}, \cdots, \mathbf{x}^{(P)}$ naturally results in an efficient variance reduction such that

**Corollary 1 (Asymptotic Normality for ICSGLD)** *Assume the same assumptions. For ICSGLD with $P$ interacting chains, we have the following weak convergence*

$$\omega_k^{-1/2}(\boldsymbol{\theta}_k^P - \widehat{\boldsymbol{\theta}}_\star) \Rightarrow \mathcal{N}(0, \boldsymbol{\Sigma}/P).$$

That is, under a similar computational budget, we have $\frac{\|\text{Var}(\boldsymbol{\theta}_{kP} - \widehat{\boldsymbol{\theta}}_\star)\|_{\text{F}}}{\|\text{Var}(\boldsymbol{\theta}_k^P - \widehat{\boldsymbol{\theta}}_\star)\|_{\text{F}}} = \frac{w_{kP}}{w_k/P} \approx P^{1-\alpha}$.

**Corollary 2 (Efficiency)** *Given a decreasing step size $\omega_k = \mathcal{O}(k^{-\alpha})$, where $0.5 < \alpha < 1$, ICSGLD is asymptotically more efficient than the single-chain CSGLD with an equivalent training cost.*

In practice, slowly decreasing step sizes are often preferred in stochastic algorithms for a better non-asymptotic performance (Benveniste et al., 1990).

## 5 EXPERIMENTS

### 5.1 LANDSCAPE EXPLORATION ON MNIST VIA THE SCALABLE RANDOM-FIELD FUNCTION

This section shows how the novel random-field function (8) facilitates the exploration of multiple modes on the MNIST dataset[§], while the standard methods, such as stochastic gradient descent (SGD) and SGLD, only *get stuck in few local modes*. To simplify the experiments, we choose a large batch size of 2500 and only pick the first five classes, namely digits from 0 to 4. The *learning rate is fixed* to 1e-6 and the temperature is set to $0.1$ [†]. We see from Figure 2(a) that both SGD and SGLD lead to fast decreasing losses. By contrast, ICSGLD yields fluctuating losses that traverse freely between high energy and low energy regions. As the particles stick in local regions, the penalty of re-visiting these zones keeps increasing until *a negative learning rate is injected* to encourage explorations.

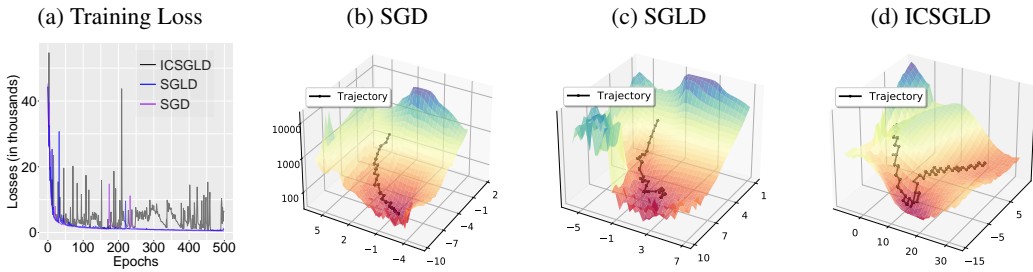

Figure 2: Visualization of mode exploration on a MNIST example based on different algorithms.

We conducted a singular value decomposition (SVD) based on the first two coordinates to visualize the trajectories: We first choose a domain that includes all the coordinates, then we recover the parameter based on the grid point and truncated values in other dimensions, and finally we fine-tune the parameters and present the approximate losses of the trajectories in Figure 2(b-d). We see SGD trajectories get stuck in a local region; SGLD *exploits a larger region* but is still quite limited in the exploration; ICSGLD, instead, first converges to a local region and then *escapes it once it over-visits this region*. This shows the strength of ICSGLD in the simulations of complex multi-modal distributions. More experimental details are presented in section D.1 of the supplementary material.

### 5.2 SIMULATIONS OF MULTI-MODAL DISTRIBUTIONS

This section shows the acceleration effect of ICSGLD via a group of simulation experiments for a multi-modal distribution. The baselines include popular Monte Carlo methods such as CSGLD, SGLD, cyclical SGLD (cycSGLD), replica exchange SGLD (reSGLD), and the particle-based SVGD.

The target multi-modal density is presented in Figure 3(a). Figure 3(b-g) displays the empirical performance of all the testing methods: the vanilla SGLD with 5 parallel chains ($\times$P5) undoubtedly

---

[§]The random-field function (Deng et al., 2020b) requires an extra perturbation term as discussed in section D4 in the supplementary material (Deng et al., 2020b); therefore it is not practically appealing in big data.

[†]Data augmentation implicitly leads to a more concentrated posterior (Wenzel et al., 2020; Aitchison, 2021).

performs the worst in this example and fails to quantify the weights of each mode correctly; the single-chain cycSGLD with 5 times of iterations (×T5) improves the performance but is still not accurate enough; reSGLD (×P5) and SVGD (×P5) have good performances, while the latter is quite costly in computations; ICSGLD (×P5) does not only traverse freely over the rugged energy landscape, but also yields the most accurate approximation to the ground truth distribution. By contrast, CSGLD (×T5) performs worse than ICSGLD and overestimates the weights on the left side. For the detailed setups, the study of convergence speed, and runtime analysis, we refer interested readers to section D.2 in the supplementary material.

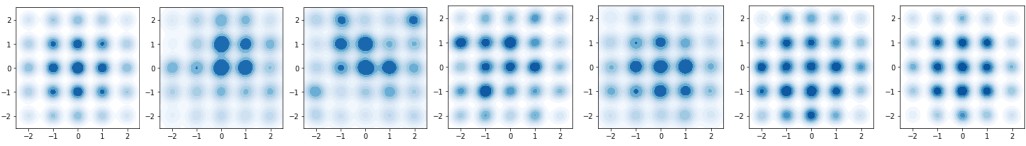

|          |          |          |        |          |          |          |
| :------: | :------: | :------: | :----: | :------: | :------: | :------: |
| (a) Truth | (b) SGLD | (c) cycSGLD | (d) SVGD | (e) reSGLD | (f) CSGLD | (g) ICSGLD |

Figure 3: Empirical behavior on a simulation dataset. Figure 3(c) and 3(f) show the simulation based on a single chain with 5 times of iterations (×T5) and the others run 5 parallel chains (×P5).

## 5.3 DEEP CONTEXTUAL BANDITS ON MUSHROOM TASKS

This section evaluates ICSGLD on the contextual bandit problem based on the UCI Mushroom data set as in Riquelme et al. (2018). The mushrooms are assumed to arrive sequentially and the agent needs to take an action at each time step based on past feedbacks. Our goal is to minimize the cumulative regret that measures the difference between the cumulative reward obtained by the proposed policy and optimal policy. We evaluate Thompson Sampling (TS) based on a variety of approximate inference methods for posterior sampling. We choose one $\epsilon$-greedy policy (EpsGreedy) based on the RMSProp optimizer with a decaying learning rate (Riquelme et al., 2018) as a baseline. Two variational methods, namely stochastic gradient descent with a constant learning rate (ConstSGD) (Mandt et al., 2017) and Monte Carlo Dropout (Dropout) (Gal & Ghahramani, 2016) are compared to approximate the posterior distribution. For the sampling algorithms, we include preconditioned SGLD (pSGLD) (Li et al., 2016), preconditioned CSGLD (pCSGLD) (Deng et al., 2020b), and preconditioned ICSGLD (pICSGLD). Note that all the algorithms run 4 parallel chains with average outputs (×P4) except that pCSGLD runs a single-chain with 4 times of computational budget (×T4). For more details, we refer readers to section D.3 in the supplementary material.

Figure 4 shows that EpsGreedy ×P4 tends to explore too much for a long horizon as expected; ConstSGD×P4 and Dropout×P4 perform poorly in the beginning but eventually outperform Eps-Greedy ×P4 due to the inclusion of uncertainty for exploration, whereas the uncertainty seems to be inadequate due to the nature of variational inference. By contrast, pSGLD×P4 significantly

outperforms the variational methods by considering preconditioners within an exact sampling framework (SGLD). As a unique algorithm that runs in a single-chain manner, pCSGLD×T4 leads to the worst performance due to the inefficiency in learning the self-adapting parameters, fortunately, pCSGLD×T4 slightly outperform pSGLD×P4 in the later phase with the help of the well-estimated self-adapting parameters. Nevertheless, pICSGLD×P4 propose to optimize the shared self-adapting parameters at the same time, which in turn greatly contributes to the simulation of the posterior. As a result, pICSGLD×P4 consistently shows the lowest regret excluding the very early period. This shows the great superiority of the interaction mechanism in learning the self-adapting parameters for accelerating the simulations.

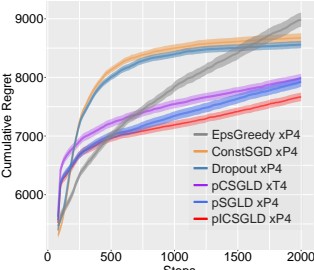

Figure 4: Cumulative regret on the mushroom task.

## 5.4 UNCERTAINTY ESTIMATION

This section evaluates the qualify of our algorithm in uncertainty quantification. For model architectures, we use residual networks (ResNet) (He et al., 2016) and a wide ResNet (WRN) (Zagoruyko & Komodakis, 2016); we choose 20, 32, and 56-layer ResNets (denoted by ResNet20, et al.) and a WRN-16-8 network, a 16-layer WRN that is 8 times wider than ResNet16. We train the models on

CIFAR100, and report the test accuracy (ACC) and test negative log-likelihood (NLL) based on 5 trials with standard error. For the out-of-distribution prediction performance, we test the well-trained models in Brier scores (Brier) * on the Street View House Numbers dataset (SVHN).

Due to the wide adoption of momentum stochastic gradient descent (M-SGD), we use stochastic gradient Hamiltonian Monte Carlo (SGHMC) (Chen et al., 2014) as the baseline sampling algorithm and denote the interacting contour SGHMC by ICSHMC. In addition, we include several high performing baselines, such as SGHMC with cyclical learning rates (cycSGHMC) (Zhang et al., 2020b), SWAG based on cyclic learning rates of 10 cycles (cycSWAG) (Maddox et al., 2019) and variance-reduced replica exchange SGHMC (reSGHMC) (Deng et al., 2021a). For a fair comparison, ICSGLD also conducts variance reduction on the energy function to alleviate the bias. Moreover, a large $\zeta = 3 \times 10^6$ is selected, which only induces mild gradient multipliers ranging from $-1$ to $2$ to penalize over-visited partitions. We don't include SVGD (Liu & Wang, 2016) and SPOS (Zhang et al., 2020a) for scalability reasons. A batch size of 256 is selected. We run 4 parallel processes ($\times$P4) with 500 epochs for M-SGD, reSGHMC and ICSGHMC and run cycSGHMC and cycSWAG 2000 epochs ($\times$T4) based on a single process with 10 cycles. Refer to section D.4 of the supplementary material for the detailed settings.

TABLE 1: UNCERTAINTY ESTIMATIONS ON CIFAR100 AND SVHN.

| MODEL | ResNet20 | | | ResNet32 | | |
|---|---|---|---|---|---|---|
| | ACC (%) | NLL | Brier (‰) | ACC (%) | NLL | Brier (‰) |
| cycSGHMC×T4 | 75.41±0.10 | 8437±30 | 2.91±0.13 | 77.93±0.17 | 7658±19 | 3.29±0.13 |
| cycSWAG×T4 | 75.46±0.11 | 8419±26 | 2.78±0.12 | 77.91±0.15 | 7656±22 | 3.19±0.14 |
| M-SGD×P4 | 76.01±0.12 | 8175±25 | 2.58±0.08 | 78.41±0.12 | 7501±23 | 2.77±0.15 |
| reSGHMC×P4 | 76.15±0.16 | 8196±27 | 2.73±0.10 | 78.57±0.07 | 7454±15 | 3.04±0.09 |
| ICSGHMC×P4 | **76.34±0.15** | **8076±31** | **2.54±0.14** | **78.72±0.16** | **7406±29** | **2.76±0.15** |

| MODEL | ResNet56 | | | WRN-16-8 | | |
|---|---|---|---|---|---|---|
| | ACC (%) | NLL | Brier (‰) | ACC (%) | NLL | Brier (‰) |
| cycSGHMC×T4 | 81.23±0.19 | 6770±59 | 3.18±0.08 | 82.98±0.03 | 6384±11 | 2.17±0.05 |
| cycSWAG×T4 | 81.14±0.11 | 6744±55 | 3.06±0.09 | 83.05±0.04 | 6359±14 | 2.04±0.07 |
| M-SGD×P4 | 81.03±0.14 | 6847±22 | **2.86±0.08** | 82.57±0.07 | 6821±21 | **1.77±0.06** |
| reSGHMC×P4 | 81.11±0.16 | 6915±40 | 2.92±0.12 | 82.72±0.08 | 6452±19 | 1.92±0.04 |
| ICSGHMC×P4 | **81.51±0.18** | **6630±38** | 2.88±0.09 | **83.12±0.10** | **6338±36** | 1.83±0.06 |

Table 1 shows that the vanilla ensemble results via M-SGD×P4 surprisingly outperform cycSGHMC×T4 and cycSWAG×T4 on medium models, such as ResNet20 and ResNet32, and show very good performance on the out-of-distribution samples in Brier scores. We suspect that the parallel implementation (×P4) provides isolated initializations with less correlated samples; by contrast, cycSGHMC×T4 and cycSWAG×T4 explore the energy landscape contiguously, implying a risk to stay near the original region. reSGHMC×P4 shows a remarkable performance overall, but demonstrates a large variance occasionally; this indicates the insufficiency of the swaps when multiple processes are included. When it comes to testing WRN-16-8, cycSWAG×T4 shows a marvelous result and a large improvement compared to the other baselines. We conjecture that cycSWAG is more independent of hyperparameter tuning, thus leading to better performance in larger models. We don't report CSGHMC×P4 since it becomes quite unstable during the training of ResNet56 and WRN-16-8 models and causes mediocre results. As to ICSGHMC×P4, it consistently performs remarkable in both ACC and NLL and performs comparable to M-SGD×P4 in Brier scores.

Code is available at github.com/WayneDW/Interacting-Contour-Stochastic-Gradient-Langevin-Dynamics.

## 6 CONCLUSION

We have proposed the ICSGLD as an efficient algorithm for sampling from distributions with a complex energy landscape, and shown theoretically that ICSGLD is indeed more efficient than the single-chain CSGLD for a slowly decreasing step size. To our best knowledge, this is the first interacting importance sampling algorithm that adapts to big data problems without scalability concerns. ICSGLD has been compared with numerous state-of-the-art baselines for various tasks, whose remarkable results indicate its promising future in big data applications.

---

*The Brier score measures the mean squared error between the predictive and actual probabilities.

## ACKNOWLEDGMENT

Liang's research was supported in part by the grants DMS-2015498, R01-GM117597 and R01-GM126089. Lin acknowledges the support from NSF (DMS-1555072, DMS-2053746, and DMS-2134209), BNL Subcontract 382247, and DE-SC0021142.

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

We summarize the supplementary material as follows: Section A provides the preliminary knowledge for stochastic approximation; Section B shows a local stability condition that adapts to high losses; Section C proves the main asymptotic normality for the stochastic approximation process, which naturally yields the conclusion that interacting contour stochastic gradient Langevin dynamics (ICSGLD) is more efficient than the analogous single chain based on slowly decreasing step sizes; Section D details the experimental settings.

# A  PRELIMINARIES

## A.1  STOCHASTIC APPROXIMATION

Given a random-field function $\widetilde{H}(\boldsymbol{\theta}, \mathbf{x})$, the stochastic approximation algorithm (Benveniste et al., 1990) proposes to solve the mean-field equation $h(\boldsymbol{\theta}) = 0$ in the analysis of adaptive algorithms

$$h(\boldsymbol{\theta}) = \int_{\mathcal{X}} \widetilde{H}(\boldsymbol{\theta}, \mathbf{x}) \varpi_{\boldsymbol{\theta}}(d\mathbf{x}) = 0,$$

where $\mathbf{x} \in \mathcal{X} \subset \mathbb{R}^d$, $\boldsymbol{\theta} \in \boldsymbol{\Theta} \subset \mathbb{R}^m$, $\varpi_{\boldsymbol{\theta}}(\mathbf{x})$ is a distribution that depends on the self-adapting parameter $\boldsymbol{\theta}$. Given the transition kernel $\Pi_{\boldsymbol{\theta}}(\boldsymbol{x}, A)$ for any Borel subset $A \subset \mathcal{X}$, the algorithm can be written as follows

(1) Simulate $\mathbf{x}_{k+1} \sim \Pi_{\boldsymbol{\theta}_k}(\mathbf{x}_k, \cdot)$, which yields the invariant distribution $\varpi_{\boldsymbol{\theta}_k}(\cdot)$,

(2) Optimize $\boldsymbol{\theta}_{k+1} = \boldsymbol{\theta}_k + \omega_{k+1} \widetilde{H}(\boldsymbol{\theta}_k, \mathbf{x}_{k+1})$.

Compared with the standard Robbins–Monro algorithm (Robbins & Monro, 1951), the algorithm proposes to simulate $\mathbf{x}$ from a transition kernel $\Pi_{\boldsymbol{\theta}}(\cdot, \cdot)$ instead of the distribution $\varpi_{\boldsymbol{\theta}}(\cdot)$ directly. In other words, , $\widetilde{H}(\boldsymbol{\theta}_k, \mathbf{x}_{k+1}) - h(\boldsymbol{\theta}_k)$ is not a Martingale but rather a Markov state-dependent noise.

## A.2  POISSON'S EQUATION

In the stochastic approximation algorithm, the sequence of $\{(\mathbf{x}_k, \boldsymbol{\theta}_k)\}_{k=1}^{\infty}$ on the product space $\mathcal{X} \times \boldsymbol{\Theta}$ is generated, which is an inhomogeneous Markov chain and requires the tool of the Poisson's equation to study the convergence

$$\mu_{\boldsymbol{\theta}}(\boldsymbol{x}) - \Pi_{\boldsymbol{\theta}} \mu_{\boldsymbol{\theta}}(\boldsymbol{x}) = \widetilde{H}(\boldsymbol{\theta}, \boldsymbol{x}) - h(\boldsymbol{\theta}),$$

where $\mu_{\boldsymbol{\theta}}(\cdot)$ is a function on $\mathcal{X}$. The solution $\mu_{\boldsymbol{\theta}}(\boldsymbol{x})$ to the Poisson's equation exists and is formulated in the following form when the above series converges:

$$\mu_{\boldsymbol{\theta}}(\mathbf{x}) := \sum_{k \geq 0} \Pi_{\boldsymbol{\theta}}^k (\widetilde{H}(\boldsymbol{\theta}, \mathbf{x}) - h(\boldsymbol{\theta})),$$

where $\Pi_{\boldsymbol{\theta}}^k (\widetilde{H}(\boldsymbol{\theta}, \mathbf{x}) - h(\boldsymbol{\theta})) = \int (\widetilde{H}(\boldsymbol{\theta}, \mathbf{y}) - h(\boldsymbol{\theta})) \Pi_{\boldsymbol{\theta}}^k(\mathbf{x}, d\mathbf{y})$. To ensure such a convergence, Benveniste et al. (1990) made the following regularity conditions on the solution $\mu_{\boldsymbol{\theta}}(\cdot)$ of the Poisson's equation:

*There exist a Lyapunov function $V : \mathcal{X} \to [1, \infty)$ and a positive constant $C > 0$ such that $\forall \boldsymbol{\theta}, \boldsymbol{\theta}' \in \boldsymbol{\Theta}$,* we have

$$\|\Pi_{\boldsymbol{\theta}} \mu_{\boldsymbol{\theta}}(\mathbf{x})\| \leq CV(\mathbf{x}), \quad \|\Pi_{\boldsymbol{\theta}} \mu_{\boldsymbol{\theta}}(\mathbf{x}) - \Pi_{\boldsymbol{\theta}'} \mu_{\boldsymbol{\theta}'}(\mathbf{x})\| \leq C\|\boldsymbol{\theta} - \boldsymbol{\theta}'\| V(\mathbf{x}), \quad \mathbb{E}[V(\mathbf{x})] \leq \infty, \quad (12)$$

where a common choice for the Lyapunov function is to set $V(\mathbf{x}) = 1 + \|\mathbf{x}\|^2$ (Teh et al., 2016; Vollmer et al., 2016).

## A.3  GAUSSIAN DIFFUSIONS

Consider a stochastic linear differential equation

$$d\boldsymbol{U}_t = h_{\boldsymbol{\theta}}(\boldsymbol{\theta}_t) \boldsymbol{U}_t dt + \boldsymbol{R}^{1/2}(\boldsymbol{\theta}_t) d\boldsymbol{W}_t, \quad (13)$$

where $\boldsymbol{U}$ is a $m$-dimensional random vector, $h_{\boldsymbol{\theta}} := \frac{d}{d\boldsymbol{\theta}} h(\boldsymbol{\theta})$, $\boldsymbol{R}(\boldsymbol{\theta}) := \sum_{k=-\infty}^{\infty} \mathrm{Cov}_{\boldsymbol{\theta}}(H(\boldsymbol{\theta}, \mathbf{x}_k), H(\boldsymbol{\theta}, \mathbf{x}_0))$ is a positive definite matrix that depends on $\boldsymbol{\theta}(\cdot)$, $\boldsymbol{W} \in \mathbb{R}^m$

is a standard Brownian motion. Given a large enough $t$ such that $\boldsymbol{\theta}_t$ converges to a fixed point $\widehat{\boldsymbol{\theta}}_\star$ sufficiently fast, we may write the diffusion associated with Eq.(13) as follow

$$\boldsymbol{U}_t \approx e^{-th\boldsymbol{\theta}(\widehat{\boldsymbol{\theta}}_\star)}\boldsymbol{U}_0 + \int_0^t e^{-(t-s)h\boldsymbol{\theta}(\widehat{\boldsymbol{\theta}}_\star)} \circ \boldsymbol{R}(\widehat{\boldsymbol{\theta}}_\star)d\boldsymbol{W}_s, \tag{14}$$

Suppose that the matrix $h_{\boldsymbol{\theta}}(\widehat{\boldsymbol{\theta}}_\star)$ is negative definite, then $\boldsymbol{U}_t$ converges in distribution to a Gaussian variable

$$\mathbb{E}[\boldsymbol{U}_t] = e^{-th\boldsymbol{\theta}(\widehat{\boldsymbol{\theta}}_\star)}\boldsymbol{U}_0$$

$$\mathrm{Var}(\boldsymbol{U}_t) = \int_0^t e^{th\boldsymbol{\theta}(\widehat{\boldsymbol{\theta}}_\star)} \circ \boldsymbol{R} \circ e^{th\boldsymbol{\theta}(\widehat{\boldsymbol{\theta}}_\star)}du.$$

The main goal of this supplementary file is to study the Gaussian approximation of the process $\omega_k^{-1/2}(\boldsymbol{\theta}_k - \widehat{\boldsymbol{\theta}}_\star)$ to the solution Eq.(14) for a proper step size $\omega_k$. Thereafter, the advantage of interacting mechanisms can be naturally derived.

## B  STABILITY AND CONVERGENCE ANALYSIS

As required by the algorithm, we update $P$ contour stochastic gradient Langevin dynamics (CSGLD) simultaneously. For the notations, we denote the particle of the p-th chain at iteration $k$ by $\mathbf{x}_k^{(p)} \in \mathcal{X} \subset \mathbb{R}^d$ and the joint state of the $P$ parallel particles at iteration $k$ by $\mathbf{x}_k^{\otimes P} := \left(\mathbf{x}_k^{(1)}, \mathbf{x}_k^{(2)}, \cdots, \mathbf{x}_k^{(P)}\right)^\top \in \mathcal{X}^{\otimes P} \subset \mathbb{R}^{dP}$. We also denote the learning rate and step size at iteration $k$ by $\epsilon_k$ and $\omega_k$, respectively. We denote by $\mathcal{N}(0, \boldsymbol{I}_{dP})$ a standard $dP$-dimensional Gaussian vector and denote by $\zeta$ a positive hyperparameter.

### B.1  ICSGLD ALGORITHM

First, we introduce the interacting contour stochastic gradient Langevin dynamics (ICSGLD) with $P$ parallel chains:

(1) Simulate $\mathbf{x}_{k+1}^{\otimes P} = \mathbf{x}_k^{\otimes P} - \epsilon_k \nabla_\mathbf{x}\widetilde{\boldsymbol{L}}(\mathbf{x}_k^{\otimes P}, \boldsymbol{\theta}_k) + \mathcal{N}(0, 2\epsilon_k\tau\boldsymbol{I}_{dP})$,      (S$_1$)

(2) Optimize $\boldsymbol{\theta}_{k+1} = \boldsymbol{\theta}_k + \omega_{k+1}\widetilde{\boldsymbol{H}}(\boldsymbol{\theta}_k, \mathbf{x}_{k+1}^{\otimes P})$,      (S$_2$)

where $\nabla_\mathbf{x}\widetilde{\boldsymbol{L}}(\mathbf{x}^{\otimes P}, \boldsymbol{\theta}) := \left(\nabla_\mathbf{x}\widetilde{L}(\mathbf{x}^{(1)}, \boldsymbol{\theta}), \nabla_\mathbf{x}\widetilde{L}(\mathbf{x}^{(2)}, \boldsymbol{\theta}), \cdots, \nabla_\mathbf{x}\widetilde{L}(\mathbf{x}^{(P)}, \boldsymbol{\theta})\right)^\top$, $\nabla_\mathbf{x}\widetilde{L}(\mathbf{x}, \boldsymbol{\theta})$ is the stochastic adaptive gradient given by

$$\nabla_\mathbf{x}\widetilde{L}(\mathbf{x}, \boldsymbol{\theta}) = \frac{N}{n}\underbrace{\left[1 + \frac{\zeta\tau}{\Delta u}\left(\log\theta(J_{\widetilde{U}}(\mathbf{x})) - \log\theta((J_{\widetilde{U}}(\mathbf{x}) - 1) \vee 1)\right)\right]}_{\text{gradient multiplier}}\nabla_\mathbf{x}\widetilde{U}(\mathbf{x}). \tag{15}$$

In particular, the interacting random-field function is written as

$$\widetilde{\boldsymbol{H}}(\boldsymbol{\theta}_k, \mathbf{x}_{k+1}^{\otimes P}) = \frac{1}{P}\sum_{p=1}^P \widetilde{H}(\boldsymbol{\theta}_k, \mathbf{x}_{k+1}^{(p)}), \tag{16}$$

where each random-field function $\widetilde{H}(\boldsymbol{\theta}, \mathbf{x}) = (\widetilde{H}_1(\boldsymbol{\theta}, \mathbf{x}), \ldots, \widetilde{H}_m(\boldsymbol{\theta}, \mathbf{x}))$ follows

$$\widetilde{H}_i(\boldsymbol{\theta}, \mathbf{x}) = \theta(J_{\widetilde{U}}(\mathbf{x}))\left(1_{i=J_{\widetilde{U}}(\mathbf{x})} - \theta(i)\right), \quad i = 1, 2, \ldots, m. \tag{17}$$

Here $J_{\widetilde{U}}(\mathbf{x})$ denotes the index $i \in \{1, 2, 3, \cdots, m\}$ such that $u_{i-1} < \frac{N}{n}\widetilde{U}(\mathbf{x}) \leq u_i$ for a set of energy partitions $\{u_i\}_{i=0}^m$ and $\widetilde{U}(\mathbf{x}) = \sum_{i \in B} U_i(\mathbf{x})$ where $U_i$ denotes the negative log of a posterior based on a single data point $i$ and $B$ denotes a mini-batch of data of size $n$. Note that the stochastic

energy estimator $\widetilde{U}(\mathbf{x})$ results in a biased estimation for the partition index $J_{\widetilde{U}}(\mathbf{x})$ due to a non-linear transformation. To avoid such a bias asymptotically with respect to the learning rate $\epsilon_k$, we may consider a variance-reduced energy estimator $\widetilde{U}_{\mathrm{VR}}(\mathbf{x})$ following Deng et al. (2021a)

$$\frac{N}{n}\widetilde{U}_{\mathrm{VR}}(\mathbf{x}) = \frac{N}{n}\sum_{i \in B_k}\left(U_i(\mathbf{x}) - U_i\left(\mathbf{x}_{q\lfloor\frac{k}{q}\rfloor}\right)\right) + \sum_{i=1}^{N}U_i\left(\mathbf{x}_{q\lfloor\frac{k}{q}\rfloor}\right),\tag{18}$$

where the control variate $\mathbf{x}_{q\lfloor\frac{k}{q}\rfloor}$ is updated every $q$ iterations.

Compared with the naïve parallelism of CSGLD, a key feature of the ICSGLD algorithm lies in the joint estimation of the interacting random-field function $\widetilde{\boldsymbol{H}}(\boldsymbol{\theta}, \mathbf{x}^{\otimes P})$ in Eq.(16) for the same mean-field function $h(\boldsymbol{\theta})$.

### B.1.1 DISCUSSIONS ON THE HYPERPARAMETERS

The most important hyperparameter is $\zeta$. A fine-tuned $\zeta$ usually leads to a small or even slightly negative learning rate in low energy regions to avoid local-trap problems. Theoretically, $\zeta$ affects the $L^2$ convergence rate hidden in the big-O notation in Lemma 3.

The other hyperparameters can be easily tuned. For example, the ResNet models yields the full loss ranging from 10,000 to 60,000 after warm-ups, we thus partition the sample space according to the energy into 200 subregions equally without tuning; since the optimization of SA is nearly convex, tuning $\{\omega_k\}$ is much easier than tuning $\{\epsilon_k\}$ for non-convex learning.

### B.1.2 DISCUSSIONS ON DISTRIBUTED COMPUTING AND COMMUNICATION COST

In shared-memory settings, the implementation is trivial and the details are omitted.

In distributed-memory settings: $\boldsymbol{\theta}_{k+1}$ is updated by the central node as follows:

- The $p$-th worker conducts the sampling step ($S_1$) and sends the indices $J_{\widetilde{U}(\mathbf{x}_{k+1}^{(p)})}$'s to the central node;
- The central node aggregates the indices from all worker and updates $\boldsymbol{\theta}_k$ based on ($S_2$);
- The central node sends $\boldsymbol{\theta}_{k+1}$ back to each worker.

We emphasize that we don't communicate the model parameters $\mathbf{x} \in \mathbb{R}^d$, but rather share the self-adapting parameter $\boldsymbol{\theta} \in \mathbb{R}^m$, where $m \ll d$. For example, WRN-16-8 has 11 M parameters (40 MB), while $\boldsymbol{\theta}$ can be set to dimension 200 of size 4 KB; hence, the communication cost is not a big issue. Moreover, the theoretical advantage still holds if the communication frequency is slightly reduced.

### B.1.3 SCALABILITY TO BIG DATA

Recall that the adaptive sampler follows that

$$\mathbf{x}_{k+1} = \mathbf{x}_k - \epsilon_{k+1}\frac{N}{n}\underbrace{\left[1 + \zeta\tau\frac{\log\theta_k(J_{\widetilde{U}}(\mathbf{x}_k)) - \log\theta_k((J_{\widetilde{U}}(\mathbf{x}_k) - 1)\vee 1)}{\Delta u}\right]}_{\text{gradient multiplier}}\nabla_{\mathbf{x}}\widetilde{U}(\mathbf{x}_k) + \sqrt{2\tau\epsilon_{k+1}}w_{k+1},$$

The key to the success of (I)CSGLD is to generate sufficiently strong bouncy moves (*negative* gradient multiplier) to escape local traps. To this end, $\zeta$ can be tuned to generate proper bouncy moves.

Take the CIFAR100 experiments for example:

- the self-adjusting mechanism fails if the gradient multiplier uniformly "equals" to 1 and a too small value of $\zeta = 1$ could lead to this issue;
- the self-adjusting mechanism works only if we choose a large enough $\zeta$ such as 3e6 to generate (desired) negative gradient multiplier in over-visited regions.

However, when we set $\zeta =$ 3e6, the original stochastic approximation (SA) update proposed in (Deng et al., 2020b) follows that

$$\theta_{k+1}(i) = \theta_k(i) + \omega_{k+1} \underbrace{\theta_k^\zeta(J_{\widetilde{U}}(\mathbf{x}_{k+1}))}_{\text{essentially 0 for } \zeta \gg 1} \left(1_{i = J_{\tilde{U}}(\mathbf{x}_{k+1})} - \theta_k(i)\right).$$

Since $\theta(i) < 1$ for any $i \in \{1, \cdots, m\}$, $\theta(i)^\zeta$ is essentially 0 for such a large $\zeta$, which means that **the original SA fails to optimize when $\zeta$ is large**. Therefore, the limited choices of $\zeta$ inevitably limits the scalability to big data problems. Our newly proposed SA scheme

$$\theta_{k+1}(i) = \theta_k(i) + \omega_{k+1} \underbrace{\theta_k(J_{\widetilde{U}}(\mathbf{x}_{k+1}))}_{\text{independent of } \zeta} \left(1_{i = J_{\tilde{U}}(\mathbf{x}_{k+1})} - \theta_k(i)\right)$$

is more independent of $\zeta$ and proposes to converge to a much smoother equilibrium $\theta_\infty^{1/\zeta}$ instead of $\theta_\infty$, where $\theta_\infty(i) = \int_{\chi_i} \pi(x)dx \propto \int_{\chi_i} e^{-\frac{U(x)}{\tau}}dx$ is the energy PDF. As such, despite the linear stability is sacrificed, the resulting algorithm is more scalable. For example, estimating $e^{-10,000 \times \frac{1}{\zeta}}$ is numerically much easier than $e^{-10,000}$ for a large $\zeta$ such as $10,000$, where $10,000$ can be induced by the high losses in training deep neural networks in big data.

## B.2 ASSUMPTIONS

A long-standing problem for stochastic approximation is the difficulty in establishing the stability property and a practical remedy for this problem is to study $\Theta$ on a fixed compact set.

**Assumption A1 (Compactness)** *The space $\Theta$ is compact and $\inf_\Theta \theta(i) > 0$ for any $i \in \{1, 2, \ldots, m\}$.*

For weaker assumptions, we refer readers to Theorem 3.2 (Fort et al., 2015), where a recurrence property can be proved for the Metropolis-based Wang-Landau algorithm, which eventually established that the estimates return to a desired compact set often enough.

Next, we lay out the smoothness assumption, which is standard in the convergence analysis of SGLD, see e.g. Mattingly et al. (2010), Raginsky et al. (2017) and Xu et al. (2018).

**Assumption A2 (Smoothness)** *$U(\boldsymbol{x})$ is $M$-smooth when there exists a positive constant $M$ that satisfies $\forall \mathbf{x}, \mathbf{x}' \in \mathcal{X}$,*

$$\|\nabla_\mathbf{x} U(\mathbf{x}) - \nabla_\mathbf{x} U(\mathbf{x}')\| \leq M \|\mathbf{x} - \mathbf{x}'\|. \tag{19}$$

In addition, we assume the dissipativity condition to ensure that the geometric ergodicity of the dynamical system holds. This assumption is also crucial for verifying the solution properties of the solution of Poisson's equation. Similar assumptions have been made in Mattingly et al. (2010); Raginsky et al. (2017) and Xu et al. (2018).

**Assumption A3 (Dissipativity)** *There exist constants $\tilde{m} > 0$ and $\tilde{b} \geq 0$ that satisfies $\forall \mathbf{x} \in \mathcal{X}$ and any $\boldsymbol{\theta} \in \Theta$,*

$$\langle \nabla_\mathbf{x} L(\mathbf{x}, \boldsymbol{\theta}), \mathbf{x} \rangle \leq \tilde{b} - \tilde{m} \|\mathbf{x}\|^2. \tag{20}$$

To further establish a bounded second moment on $\mathbf{x} \in \mathcal{X}$ with respect to a proper Lyapunov function $V(\mathbf{x})$, we impose the following conditions on the gradient noise:

**Assumption A4 (Gradient noise)** *The stochastic gradient based on mini-batch settings is an unbiased estimator such that*

$$\mathbb{E}[\nabla_\mathbf{x} \widetilde{U}(\mathbf{x}_k) - \nabla_\mathbf{x} U(\mathbf{x}_k)] = 0;$$

*furthermore, for some positive constants $M$ and $B$, we have*

$$\mathbb{E}[\|\nabla_\mathbf{x} \widetilde{U}(\mathbf{x}_k) - \nabla_\mathbf{x} U(\mathbf{x}_k)\|^2] \leq M^2 \|\mathbf{x}\|^2 + B^2,$$

*where $\mathbb{E}[\cdot]$ acts on the distribution of the noise in the stochastic gradient $\nabla_\mathbf{x} \widetilde{U}(\mathbf{x}_k)$.*

### B.3 LOCAL STABILITY VIA THE SCALABLE RANDOM-FIELD FUNCTION

Now, we are ready to present our first result. Lemma 3 establishes a local stability condition for the non-linear mean-field system of ICSGLD, which implies a potential convergence of $\boldsymbol{\theta}_k$ to a unique fixed point that adapts to a wide energy range under mild assumptions.

**Lemma 3 (Local stability, restatement of Lemma 1)** *Assume Assumptions A1-A4 hold. Given any small enough learning rate $\epsilon$, a large enough $m$ and batch size $n$, and any $\boldsymbol{\theta} \in \widetilde{\Theta}$, where $\widetilde{\Theta}$ is a small neighborhood of $\boldsymbol{\theta}_\star$ that contains $\widehat{\boldsymbol{\theta}}_\star$, we have $\langle h(\boldsymbol{\theta}), \boldsymbol{\theta}-\widehat{\boldsymbol{\theta}}_\star\rangle \leq -\phi\|\boldsymbol{\theta}-\widehat{\boldsymbol{\theta}}_\star\|^2$, where $\widehat{\boldsymbol{\theta}}_\star = \boldsymbol{\theta}_\star+\mathcal{O}(\varepsilon)$,*

$$\varepsilon = \mathcal{O}\left(\sup_{\mathbf{x}} \mathrm{Var}(\xi_n(\mathbf{x})) + \epsilon + \frac{1}{m}\right) \text{ and } \boldsymbol{\theta}_\star = \left(\frac{\left(\int_{\mathcal{X}_1} \pi(\mathbf{x})d\mathbf{x}\right)^{\frac{1}{\zeta}}}{\sum_{k=1}^m \left(\int_{\mathcal{X}_k} \pi(\mathbf{x})d\mathbf{x}\right)^{\frac{1}{\zeta}}}, \dots, \frac{\left(\int_{\mathcal{X}_m} \pi(\mathbf{x})d\mathbf{x}\right)^{\frac{1}{\zeta}}}{\sum_{k=1}^m \left(\int_{\mathcal{X}_k} \pi(\mathbf{x})d\mathbf{x}\right)^{\frac{1}{\zeta}}}\right),$$

*$\phi = \inf_{\boldsymbol{\theta}} \min_i \widehat{Z}_{\zeta,\theta(i)}^{-1}\left(1 - \mathcal{O}(\varepsilon)\right) > 0$, $\widehat{Z}_{\zeta,\theta(i)}$ is defined below Eq.(28), and $\xi_n(\mathbf{x})$ denotes the noise in the energy estimator $\widetilde{U}(\mathbf{x})$ of batch size $n$ and $\mathrm{Var}(\cdot)$ denotes the variance.*

**Proof** The random-field function $\widetilde{H}_i(\boldsymbol{\theta}, \mathbf{x}) = \theta(J_{\widetilde{U}}(\mathbf{x})) \left(1_{i=J_{\widetilde{U}}(\mathbf{x})} - \theta(i)\right)$ based on the stochastic energy estimator $\widetilde{U}(\mathbf{x})$ yields a biased estimator of $H_i(\boldsymbol{\theta}, \mathbf{x}) = \theta(J(\mathbf{x})) \left(1_{i=J(\mathbf{x})} - \theta(i)\right)$ for any $i \in \{1, 2, \dots, m\}$ based on the exact energy partition function $J(\cdot)$. By Lemma.4, we know that the bias caused by the stochastic energy is of order $\mathcal{O}(\mathrm{Var}(\xi_n(\mathbf{x})))$.

Now we compute the mean-field function $h(\boldsymbol{\theta})$ based on the measure $\varpi_{\boldsymbol{\theta}}(\mathbf{x})$ simulated from SGLD:

$$h_i(\boldsymbol{\theta}) = \int_{\mathcal{X}} \widetilde{H}_i(\boldsymbol{\theta}, \mathbf{x})\varpi_{\boldsymbol{\theta}}(\mathbf{x})d\mathbf{x} = \int_{\mathcal{X}} H_i(\boldsymbol{\theta}, \mathbf{x})\varpi_{\boldsymbol{\theta}}(\mathbf{x})d\mathbf{x} + \mathcal{O}\left(\mathrm{Var}(\xi_n(\mathbf{x}))\right)$$

$$= \int_{\mathcal{X}} H_i(\boldsymbol{\theta}, \mathbf{x}) \left(\underbrace{\varpi_{\widetilde{\Psi}_{\boldsymbol{\theta}}}(\mathbf{x})}_{I_1} \underbrace{-\varpi_{\widetilde{\Psi}_{\boldsymbol{\theta}}}(\mathbf{x}) + \varpi_{\Psi_{\boldsymbol{\theta}}}(\mathbf{x})}_{I_2 : \text{piece-wise approximation}} \underbrace{-\varpi_{\Psi_{\boldsymbol{\theta}}}(\mathbf{x}) + \varpi_{\boldsymbol{\theta}}(\mathbf{x})}_{I_3 : \text{numerical discretization}}\right) d\mathbf{x} + \mathcal{O}\left(\mathrm{Var}(\xi_n(\mathbf{x}))\right),$$

$$(21)$$

where $\varpi_{\boldsymbol{\theta}}$ is the invariant measure simulated via SGLD that approximates $\varpi_{\Psi_{\boldsymbol{\theta}}}(\mathbf{x})$. $\varpi_{\Psi_{\boldsymbol{\theta}}}(\mathbf{x})$ and $\varpi_{\widetilde{\Psi}_{\boldsymbol{\theta}}}(\mathbf{x})$ are two invariant measures that follow $\varpi_{\Psi_{\boldsymbol{\theta}}}(\mathbf{x}) \propto \frac{\pi(\mathbf{x})}{\Psi_{\boldsymbol{\theta}}^\zeta(U(\mathbf{x}))}$ and $\varpi_{\widetilde{\Psi}_{\boldsymbol{\theta}}}(\mathbf{x}) \propto \frac{\pi(\mathbf{x})}{\widetilde{\Psi}_{\boldsymbol{\theta}}^\zeta(U(\mathbf{x}))}$; $\Psi_{\boldsymbol{\theta}}(u)$ and $\widetilde{\Psi}_{\boldsymbol{\theta}}(u)$ are piecewise continuous and constant functions, respectively

$$\Psi_{\boldsymbol{\theta}}(u) = \sum_{k=1}^m \left(\theta(k-1)e^{(\log\theta(k)-\log\theta(k-1))\frac{u-u_{k-1}}{\Delta u}}\right) 1_{u_{k-1}<u\leq u_k}; \quad \widetilde{\Psi}_{\boldsymbol{\theta}}(u) = \sum_{k=1}^m \theta(k)1_{u_{k-1}<u\leq u_k}.$$

$$(22)$$

(i) For the first term $I_1$, we have

$$\int_{\mathcal{X}} H_i(\boldsymbol{\theta}, \mathbf{x})\varpi_{\widetilde{\Psi}_{\boldsymbol{\theta}}}(\mathbf{x})d\mathbf{x} = \frac{1}{\widetilde{Z}_{\zeta+1,\boldsymbol{\theta}}} \int_{\mathcal{X}} \theta(J(\mathbf{x})) \left(1_{i=J(\mathbf{x})} - \theta(i)\right) \frac{\pi(\mathbf{x})}{\theta^\zeta(J(\mathbf{x}))}d\mathbf{x}$$

$$= \frac{1}{\widetilde{Z}_{\zeta+1,\boldsymbol{\theta}}} \sum_{k=1}^m \int_{\mathcal{X}_k} (1_{i=k} - \theta(i)) \frac{\pi(\mathbf{x})}{\theta^{\zeta-1}(k)}d\mathbf{x}$$

$$= \frac{1}{\widetilde{Z}_{\zeta+1,\boldsymbol{\theta}}} \left[\sum_{k=1}^m \int_{\mathcal{X}_k} \frac{\pi(\mathbf{x})}{\theta^{\zeta-1}(k)}1_{k=i}d\mathbf{x} - \theta(i)\sum_{k=1}^m \int_{\mathcal{X}_k} \frac{\pi(\mathbf{x})}{\theta^{\zeta-1}(k)}d\mathbf{x}\right] \quad (23)$$

$$= \frac{1}{\widetilde{Z}_{\zeta+1,\boldsymbol{\theta}}} \left[\frac{\int_{\mathcal{X}_i} \pi(\mathbf{x})d\mathbf{x}}{\theta^{\zeta-1}(i)} - \theta(i)\widetilde{Z}_{\zeta,\boldsymbol{\theta}}\right],$$

where $\widetilde{Z}_{\zeta+1,\boldsymbol{\theta}} = \sum_{k=1}^m \frac{\int_{\mathcal{X}_k} \pi(\mathbf{x})d\mathbf{x}}{\theta^\zeta(k)}$ denotes the normalizing constant of $\varpi_{\widetilde{\Psi}_{\boldsymbol{\theta}}}(\mathbf{x})$.

The solution $\boldsymbol{\theta}_\star$ that solves $\frac{\int_{\mathcal{X}_k} \pi(\mathbf{x})d\mathbf{x}}{\theta^{\zeta-1}(k)} - \theta(k)\widetilde{Z}_{\zeta,\boldsymbol{\theta}} = 0$ for any $k \in \{1, 2, \cdots, m\}$ satisfies $\theta_\star(k) = \left(\frac{\int_{\mathcal{X}_k} \pi(\mathbf{x})d\mathbf{x}}{\widetilde{Z}_{\zeta,\boldsymbol{\theta}_\star}}\right)^{\frac{1}{\zeta}}$. Combining the definition of $\widetilde{Z}_{\zeta,\boldsymbol{\theta}_\star} = \sum_{k=1}^m \frac{\int_{\mathcal{X}_k} \pi(\mathbf{x})d\mathbf{x}}{\theta^{\zeta-1}_\star(k)}$, we have

$$\widetilde{Z}_{\zeta,\boldsymbol{\theta}_\star} = \sum_{k=1}^m \frac{\int_{\mathcal{X}_k} \pi(\mathbf{x})d\mathbf{x}}{\theta^{\zeta-1}_\star(k)} = \sum_{k=1}^m \frac{\int_{\mathcal{X}_k} \pi(\mathbf{x})d\mathbf{x}}{\left(\frac{\int_{\mathcal{X}_k} \pi(\mathbf{x})d\mathbf{x}}{\widetilde{Z}_{\zeta,\boldsymbol{\theta}_\star}}\right)^{\frac{\zeta-1}{\zeta}}}$$

$$= \widetilde{Z}_{\zeta,\boldsymbol{\theta}_\star}^{\frac{\zeta-1}{\zeta}} \sum_{k=1}^m \frac{\int_{\mathcal{X}_k} \pi(\mathbf{x})d\mathbf{x}}{\left(\int_{\mathcal{X}_k} \pi(\mathbf{x})d\mathbf{x}\right)^{\frac{\zeta-1}{\zeta}}} = \widetilde{Z}_{\zeta,\boldsymbol{\theta}_\star}^{\frac{\zeta-1}{\zeta}} \sum_{k=1}^m \left(\int_{\mathcal{X}_k} \pi(\mathbf{x})d\mathbf{x}\right)^{\frac{1}{\zeta}},$$

which leads to $\widetilde{Z}_{\zeta,\boldsymbol{\theta}_\star} = \left(\sum_{k=1}^m \left(\int_{\mathcal{X}_k} \pi(\mathbf{x})d\mathbf{x}\right)^{\frac{1}{\zeta}}\right)^\zeta$. In other words, the mean-field system without perturbations yields a unique solution $\theta_\star(i) = \frac{\left(\int_{\mathcal{X}_i} \pi(\mathbf{x})d\mathbf{x}\right)^{\frac{1}{\zeta}}}{\sum_{k=1}^m \left(\int_{\mathcal{X}_k} \pi(\mathbf{x})d\mathbf{x}\right)^{\frac{1}{\zeta}}}$ for any $i \in \{1, 2, \cdots, m\}$.

(ii) For the second term $\mathrm{I}_2$, we have

$$\int_{\mathcal{X}} H_i(\boldsymbol{\theta}, \mathbf{x})(-\varpi_{\widetilde{\Psi}_{\boldsymbol{\theta}}}(\mathbf{x}) + \varpi_{\Psi_{\boldsymbol{\theta}}}(\mathbf{x}))d\mathbf{x} = \mathcal{O}\left(\frac{1}{m}\right), \tag{24}$$

where the result follows from the boundedness of $H(\boldsymbol{\theta}, \mathbf{x})$ in (A1) and Lemma B4 (Deng et al., 2020b).

(iii) For the last term $\mathrm{I}_3$, following Theorem 6 of Sato & Nakagawa (2014), we have for any fixed $\boldsymbol{\theta}$,

$$\int_{\mathcal{X}} H_i(\boldsymbol{\theta}, \mathbf{x})\left(-\varpi_{\Psi_{\boldsymbol{\theta}}}(\mathbf{x}) + \varpi_{\boldsymbol{\theta}}(\mathbf{x})\right)d\mathbf{x} = \mathcal{O}(\epsilon). \tag{25}$$

Plugging Eq.(23), Eq.(24) and Eq.(25) into Eq.(21), we have

$$h_i(\boldsymbol{\theta}) = \widetilde{Z}_{\zeta+1,\boldsymbol{\theta}}^{-1}\left[\varepsilon\tilde{\beta}_i(\boldsymbol{\theta}) + \frac{\int_{\mathcal{X}_i} \pi(\mathbf{x})d\mathbf{x}}{\theta^{\zeta-1}(i)} - \theta(i)\widetilde{Z}_{\zeta,\boldsymbol{\theta}}\right]$$

$$= \widetilde{Z}_{\zeta+1,\boldsymbol{\theta}}^{-1}\frac{\widetilde{Z}_{\zeta,\boldsymbol{\theta}_\star}}{\theta^{\zeta-1}(i)}\left[\varepsilon\tilde{\beta}_i(\boldsymbol{\theta})\frac{\theta^{\zeta-1}(i)}{\widetilde{Z}_{\zeta,\boldsymbol{\theta}_\star}} + \frac{\int_{\mathcal{X}_i} \pi(\mathbf{x})d\mathbf{x}}{\widetilde{Z}_{\zeta,\boldsymbol{\theta}_\star}} - \theta^\zeta(i)\frac{\widetilde{Z}_{\zeta,\boldsymbol{\theta}}}{\widetilde{Z}_{\zeta,\boldsymbol{\theta}_\star}}\right] \tag{26}$$

$$= \widetilde{Z}_{\zeta+1,\boldsymbol{\theta}}^{-1}\frac{\widetilde{Z}_{\zeta,\boldsymbol{\theta}_\star}}{\theta^{\zeta-1}(i)}\left[\varepsilon\tilde{\beta}_i(\boldsymbol{\theta})\frac{\theta^{\zeta-1}(i)}{\widetilde{Z}_{\zeta,\boldsymbol{\theta}_\star}} + \theta^\zeta_\star(i) - (\theta(i)C_{\boldsymbol{\theta}})^\zeta\right],$$

where $\tilde{\beta}_i(\boldsymbol{\theta})$ is a bounded term such that $\widetilde{Z}_{\zeta+1,\boldsymbol{\theta}}^{-1}\varepsilon\tilde{\beta}_i(\boldsymbol{\theta}) = \mathcal{O}\left(\mathrm{Var}(\xi_n(\mathbf{x})) + \epsilon + \frac{1}{m}\right)$, $\varepsilon = \mathcal{O}\left(\sup_{\mathbf{x}} \mathrm{Var}(\xi_n(\mathbf{x})) + \epsilon + \frac{1}{m}\right)$ and $C_{\boldsymbol{\theta}} = \left(\frac{\widetilde{Z}_{\zeta,\boldsymbol{\theta}}}{\widetilde{Z}_{\zeta,\boldsymbol{\theta}_\star}}\right)^{\frac{1}{\zeta}}$. By the definition of $\widetilde{Z}_{\zeta,\boldsymbol{\theta}} = \sum_{k=1}^m \frac{\int_{\mathcal{X}_k} \pi(\mathbf{x})d\mathbf{x}}{\theta^{\zeta-1}(k)}$, when $\zeta = 1$, $C_{\boldsymbol{\theta}} \equiv 1$ for any $\boldsymbol{\theta} \in \boldsymbol{\Theta}$, which suggests that the stability condition doesn't rely on the initialization of $\boldsymbol{\theta}$; however, when $\zeta \neq 1$, $C_{\boldsymbol{\theta}} \neq 1$ when $\boldsymbol{\theta} \neq \boldsymbol{\theta}_\star$, we see that $h_i(\boldsymbol{\theta}) \propto \theta_\star(i)^\zeta - (\theta(i)C_{\boldsymbol{\theta}})^\zeta +$ perturbations is a non-linear mean-field system and requires a proper initialization of $\boldsymbol{\theta} \in \widetilde{\boldsymbol{\Theta}}$.

For any $\boldsymbol{\theta} \in \widetilde{\boldsymbol{\Theta}} \subset \boldsymbol{\Theta}$ being close enough to $\boldsymbol{\theta}_\star$, there exists a Lipschitz constant $L_{\widetilde{\boldsymbol{\theta}}} = \sup_{i \leq m, \boldsymbol{\theta} \in \widetilde{\boldsymbol{\Theta}}} \frac{|C_{\boldsymbol{\theta}_\star} - C_{\boldsymbol{\theta}}|}{|\theta_\star(i) - \theta(i)|} < \infty$. By $C_{\boldsymbol{\theta}_\star} = 1$, $\theta(i) \leq 1$, and mean value theorem for some $\widetilde{\theta}(i) \in [\theta(i), \theta_\star(i)]$, we have

$$|\theta^\zeta_\star(i) - (\theta(i)C_{\boldsymbol{\theta}})^\zeta| = \zeta(\widetilde{\theta}(i)C_{\widetilde{\boldsymbol{\theta}}})^{\zeta-1}|\theta_\star(i) - \theta(i)C_{\boldsymbol{\theta}}|$$

$$= \zeta(\widetilde{\theta}(i)C_{\widetilde{\boldsymbol{\theta}}})^{\zeta-1}|\theta_\star(i) - \theta(i) + \theta(i)C_{\boldsymbol{\theta}_\star} - \theta(i)C_{\boldsymbol{\theta}}|$$

$$\leq \zeta(\widetilde{\theta}(i)C_{\widetilde{\boldsymbol{\theta}}})^{\zeta-1}|\theta_\star(i) - \theta(i)| + \theta(i)|C_{\boldsymbol{\theta}_\star} - C_{\boldsymbol{\theta}}|$$

$$\leq \zeta(\widetilde{\theta}(i)C_{\widetilde{\boldsymbol{\theta}}})^{\zeta-1}(1 + L_{\widetilde{\boldsymbol{\theta}}})|\theta_\star(i) - \theta(i)|, \tag{27}$$

Combining Eq.(26) and Eq.(27), we have

$$
h_i(\boldsymbol{\theta}) = \widetilde{Z}_{\zeta+1,\boldsymbol{\theta}}^{-1} \frac{\widetilde{Z}_{\zeta,\boldsymbol{\theta}_\star}}{\theta^{\zeta-1}(i)} \left[ \varepsilon \tilde{\beta}_i(\boldsymbol{\theta}) \frac{\theta^{\zeta-1}(i)}{\widetilde{Z}_{\zeta,\boldsymbol{\theta}_\star}} + \theta_\star^\zeta(i) - (\theta(i)C_{\boldsymbol{\theta}})^\zeta \right]
$$

$$
= \widehat{Z}_{\zeta,\theta(i)}^{-1} \left[ \varepsilon \beta_i(\boldsymbol{\theta}) + \theta_\star(i) - \theta(i) \right],
\tag{28}
$$

where $\widehat{Z}_{\zeta,\theta(i)}^{-1} = \frac{\widetilde{Z}_{\zeta+1,\boldsymbol{\theta}}^{-1}\widetilde{Z}_{\zeta,\boldsymbol{\theta}_\star}}{\zeta(\widetilde{\theta}(i)C_{\widetilde{\boldsymbol{\theta}}})^{\zeta-1}(1+L_{\widetilde{\boldsymbol{\theta}}})\theta^{\zeta-1}(i)}$; $\beta_i(\boldsymbol{\theta})$ is some bounded term such that $\beta_i(\boldsymbol{\theta}) \leq$

$\frac{\tilde{\beta}_i(\boldsymbol{\theta})\theta^{\zeta-1}(i)}{\zeta(\widetilde{\theta}(i)C_{\widehat{\boldsymbol{\theta}}})^{\zeta-1}(1+L_{\widehat{\boldsymbol{\theta}}})\widetilde{Z}_{\zeta,\boldsymbol{\theta}_\star}}$; $C_{\widetilde{\boldsymbol{\theta}}} = \left( \frac{\widetilde{Z}_{\zeta,\widetilde{\boldsymbol{\theta}}}}{\widetilde{Z}_{\zeta,\boldsymbol{\theta}_\star}} \right)^{\frac{1}{\zeta}}$; $L_{\widetilde{\boldsymbol{\theta}}} = \sup_{i \leq m, \boldsymbol{\theta} \in \widetilde{\Theta}} \frac{|C_{\boldsymbol{\theta}_\star} - C_{\boldsymbol{\theta}}|}{|\theta_\star(i) - \theta(i)|} < \infty.$

Next, we apply the perturbation theory to solve the ODE system with small disturbances (Weinhart et al., 2010) and obtain the equilibrium $\widehat{\boldsymbol{\theta}}_\star$,

where $\varepsilon \boldsymbol{\beta}(\widehat{\boldsymbol{\theta}}_\star) + \boldsymbol{\theta}_\star - \widehat{\boldsymbol{\theta}}_\star = 0$, to the mean-field equation $h_i(\boldsymbol{\theta})$ such that

$$
h_i(\boldsymbol{\theta}) = \widehat{Z}_{\zeta,\theta(i)}^{-1} \left[ \varepsilon \beta_i(\boldsymbol{\theta}) + \theta_\star(i) - \theta(i) \right]
$$

$$
= \widehat{Z}_{\zeta,\theta(i)}^{-1} \left[ \varepsilon \beta_i(\boldsymbol{\theta}) - \varepsilon \beta_i(\widehat{\boldsymbol{\theta}}_\star) + \varepsilon \beta_i(\widehat{\boldsymbol{\theta}}_\star) + \theta_\star(i) - \theta(i) \right]
$$

$$
= \widehat{Z}_{\zeta,\theta(i)}^{-1} \left[ \mathcal{O}(\varepsilon)(\theta(i) - \widehat{\theta}_\star(i)) + \widehat{\theta}_\star(i) - \theta(i) \right]
\tag{29}
$$

$$
= \widehat{Z}_{\zeta,\theta(i)}^{-1} \left( 1 - \mathcal{O}(\varepsilon) \right) \left( \widehat{\theta}_\star(i) - \theta(i) \right),
$$

where a smoothness condition clearly holds for the $\boldsymbol{\beta}(\cdot)$ function [†]. Given a positive definite Lyapunov function $\mathbb{V}(\boldsymbol{\theta}) = \frac{1}{2}\|\widehat{\boldsymbol{\theta}}_\star - \boldsymbol{\theta}\|^2$, the mean-field system $h(\boldsymbol{\theta}) = \widehat{Z}_{\zeta,\theta(i)}^{-1}(\varepsilon\boldsymbol{\beta}(\boldsymbol{\theta}) + \boldsymbol{\theta}_\star - \boldsymbol{\theta}) = \widehat{Z}_{\zeta,\theta(i)}^{-1}(1 - \mathcal{O}(\varepsilon))(\widehat{\boldsymbol{\theta}}_\star - \boldsymbol{\theta})$ for $i \in \{1, 2, \cdots, m\}$ enjoys the following property

$$
\langle h(\boldsymbol{\theta}), \nabla\mathbb{V}(\boldsymbol{\theta}) \rangle = \langle h(\boldsymbol{\theta}), \boldsymbol{\theta} - \widehat{\boldsymbol{\theta}}_\star \rangle
$$

$$
\leq -\min_i \widehat{Z}_{\zeta,\theta(i)}^{-1}(1 - \mathcal{O}(\varepsilon))\|\boldsymbol{\theta} - \widehat{\boldsymbol{\theta}}_\star\|^2
$$

$$
\leq -\phi\|\boldsymbol{\theta} - \widehat{\boldsymbol{\theta}}_\star\|^2,
$$

where $\phi = \inf_{\boldsymbol{\theta}} \min_i \widehat{Z}_{\zeta,\theta(i)}^{-1}(1 - \mathcal{O}(\varepsilon)) > 0$ given the compactness assumption A1 and a small enough $\varepsilon = \mathcal{O}\left(\sup_{\mathbf{x}} \text{Var}(\xi_n(\mathbf{x})) + \epsilon + \frac{1}{m}\right)$. ∎

**Remark 1** *The newly proposed random-field function Eq.(17) may sacrifice the global stability by including an approximately linear mean-field system Eq.(28) instead of a linear stable system (see formula (15) in Deng et al. (2020b)). The advantage, however, is that such a mechanism facilitates the estimation of $\boldsymbol{\theta}_\star$. We emphasize that the original energy probability in each partition $\left\{ \int_{\mathcal{X}_k} \pi(\mathbf{x})d\mathbf{x} \right\}_{k=1}^m$ (Deng et al., 2020b) may be very difficult to estimate for big data problems. By contrast, the estimation of $\left\{ \left( \int_{\mathcal{X}_k} \pi(\mathbf{x})d\mathbf{x} \right)^{\frac{1}{\zeta}} \right\}_{k=1}^m$ becomes much easier given a proper $\zeta > 0$.*

**Technical lemmas**

**Lemma 4** *The stochastic energy estimator $\widetilde{U}(\mathbf{x})$ leads to a controllable bias in the random-field function.*

$$
|\mathbb{E}[\widetilde{H}_i(\boldsymbol{\theta}, \mathbf{x})] - H_i(\boldsymbol{\theta}, \mathbf{x})| = \mathcal{O}\left( \text{Var}(\xi_n(\mathbf{x})) \right),
$$

*where the expectation $\mathbb{E}[\cdot]$ is taken with respect to the random noise in the stochastic energy estimator of $\widetilde{U}(\cdot)$.*

**Proof** Denote the noise in the stochastic energy estimator by $\xi(\mathbf{x})$, such that $\widetilde{U}(\cdot) = U(\cdot) + \xi(\cdot)$. Recall that $\widetilde{H}_i(\boldsymbol{\theta}, \mathbf{x}) = \theta(J_{\widetilde{U}}(\mathbf{x}))\left( 1_{i=J_{\widetilde{U}}(\mathbf{x})} - \theta(i) \right)$ and $J_{\widetilde{U}}(\mathbf{x}) \in \{1, 2, \cdots, m\}$ satisfies

---

[†]A small change of $\boldsymbol{\theta}$ won't significantly affect the perturbations caused by Eq.(24), Eq.(25) and $\text{Var}(\xi_n(\cdot))$.

$u_{J_{\widetilde{U}}(\mathbf{x})-1} < \frac{N}{n}\widetilde{U}(\mathbf{x}) \leq u_{J_{\widetilde{U}}(\mathbf{x})}$ for a set of energy partitions $\{u_i\}_{i=0}^{m}$. We can interpret $\widetilde{H}_i(\boldsymbol{\theta}, \mathbf{x})$ as a non-linear transformation $\Phi$ that maps $\widetilde{U}(\mathbf{x})$ to $(0, 1)$. Similarly, $H_i(\boldsymbol{\theta}, \mathbf{x})$ maps $U(\mathbf{x})$ to $(0, 1)$. In what follows, the bias of random-field function is upper bounded as follows

$$
\begin{aligned}
|\mathbb{E}[\widetilde{H}_i(\boldsymbol{\theta}, \mathbf{x})] - H_i(\boldsymbol{\theta}, \mathbf{x})| &= \left| \int \Phi(U(\mathbf{x}) + \xi(\mathbf{x})) - \Phi(U(\mathbf{x}))d\mu(\xi(\mathbf{x})) \right| \\
&= \left| \int \xi(\mathbf{x})\Phi'(U(\mathbf{x})) + \frac{\xi(\mathbf{x})^2}{2}\Phi''(u)d\mu(\xi(\mathbf{x})) \right| \\
&\leq \left| \int \xi_n(\mathbf{x})\Phi'(U(\mathbf{x}))d\mu(\xi_n(\mathbf{x})) \right| + \left| \frac{\Phi''(u)}{2}\int \xi_n(\mathbf{x})^2 d\mu(\xi_n(\mathbf{x})) \right| \\
&\leq \mathcal{O}\left(\mathrm{Var}(\xi_n(\mathbf{x}))\right),
\end{aligned}
$$

where $\mu(\xi(\mathbf{x}))$ is the probability measure associated with $\xi(\mathbf{x})$; the second equality follows from Taylor expansion for some energy $u$ and the third equality follows because the stochastic energy estimator is unbiased; $\Phi'(U(\mathbf{x})) = \mathcal{O}(\frac{\theta(J(\mathbf{x}))-\theta(J(\mathbf{x})-1)}{\Delta u})$ is clearly bounded due to the definition of $\boldsymbol{\theta}$; a similar conclusion also applies to $\Phi''(\cdot)$. The last inequality easily follows by applying Cauchy Schwarz inequality.

### B.4 CONVERGENCE OF THE SELF-ADAPTING PARAMETERS

The following is a restatement of Lemma 3.2 of Raginsky et al. (2017), which holds for any $\boldsymbol{\theta}$ in the compact space $\boldsymbol{\Theta}$.

**Lemma 5 (Uniform $L^2$ bounds)** *Assume Assumptions A1, A3 and A4 hold. We have a bounded second moment $\sup_{k\geq 1}\mathbb{E}[\|\mathbf{x}_k\|^2] < \infty$ given a small enough learning rate.*

The following lemma justifies the regularity properties of Poisson's equation, which is crucial in controlling the perturbations through the stochastic approximation process. The first version was proposed in Lemma B2 of Deng et al. (2020b). Now we give a more detailed proof by utilizing a Lyapunov function $V(\mathbf{x}) = 1 + \mathbf{x}^2$ and Lemma 5.

**Lemma 6 (Solution of Poisson's equation)** *Assume that Assumptions A1-A4 hold. There is a solution $\mu_{\boldsymbol{\theta}}(\cdot)$ on $\mathcal{X}$ to the Poisson's equation*

$$
\mu_{\boldsymbol{\theta}}(\boldsymbol{x}) - \Pi_{\boldsymbol{\theta}}\mu_{\boldsymbol{\theta}}(\boldsymbol{x}) = \widetilde{H}(\boldsymbol{\theta}, \boldsymbol{x}) - h(\boldsymbol{\theta}). \tag{30}
$$

*Furthermore, there exists a constant $C$ such that for all $\boldsymbol{\theta}, \boldsymbol{\theta}' \in \boldsymbol{\Theta}$*

$$
\begin{aligned}
\mathbb{E}[\|\Pi_{\boldsymbol{\theta}}\mu_{\boldsymbol{\theta}}(\mathbf{x})\|] &\leq C, \\
\mathbb{E}[\|\Pi_{\boldsymbol{\theta}}\mu_{\boldsymbol{\theta}}(\mathbf{x}) - \Pi_{\boldsymbol{\theta}'}\mu_{\boldsymbol{\theta}'}(\mathbf{x})\|] &\leq C\|\boldsymbol{\theta} - \boldsymbol{\theta}'\|.
\end{aligned} \tag{31}
$$

**Proof** The existence and the regularity property of Poisson's equation can be used to control the perturbations. The key of the proof lies in verifying drift conditions proposed in Section 6 of Andrieu et al. (2005).

**(DRI)** By the smoothness assumption A2, we have that $U(\mathbf{x})$ is continuously differentiable almost everywhere. By the dissipative assumption A3 and Theorem 2.1 (Roberts & Tweedie, 1996), we can show that the discrete dynamics system is irreducible and aperiodic. Now consider a Lyapunov function $V = 1 + \|\mathbf{x}\|^2$ and any compact subset $\mathcal{K} \subset \boldsymbol{\Theta}$, the drift conditions are verified as follows:

**(DRI1)** Given small enough learning rates $\{\epsilon_k\}_{k\geq 1}$, the smoothness assumption A2, and the dissipative assumption A3, applying Corollary 7.5 (Mattingly et al., 2002) yields the minorization condition for the CSGLD algorithm, i.e. there exists $\eta > 0$, a measure $\nu$, and a set $\mathcal{C}$ such that $\nu(\mathcal{C}) = 1$. Moreover, we have

$$
P_{\boldsymbol{\theta}\in\mathcal{K}}(x, A) \geq \eta\nu(A) \quad \forall A \in \mathcal{X}, \mathbf{x} \in \mathcal{C}. \tag{I}
$$

where $P_{\boldsymbol{\theta}}(\mathbf{x}, \mathbf{y}) := \frac{1}{2\sqrt{(4\pi\epsilon)^{d/2}}}\mathbb{E}\left[e^{-\frac{\|\mathbf{y}-\mathbf{x}+\epsilon\nabla_{\mathbf{x}}\widetilde{L}(\mathbf{x},\boldsymbol{\theta})\|^2}{4\epsilon}}|\mathbf{x}\right]$ denotes the transition kernel based on CS-GLD with the parameter $\boldsymbol{\theta} \in \mathcal{K}$ and a learning rate $\epsilon$, in addition, the expectation is taken over the

adaptive gradient $\nabla_{\mathbf{x}}\widetilde{L}(\mathbf{x}, \boldsymbol{\theta})$ in Eq.(15). Using Assumption A1-A4, we can prove the uniform L2 upper bound by following Lemma 3.2 (Raginsky et al., 2017). Further, by Theorem 7.2 (Mattingly et al., 2002), there exist $\tilde{\alpha} \in (0, 1)$ and $\tilde{\beta} \geq 0$ such that

$$P_{\boldsymbol{\theta} \in \mathcal{K}} V(\mathbf{x}) \leq \tilde{\alpha} V(\mathbf{x}) + \tilde{\beta}. \tag{II}$$

Consider a Lyapunov function $V = 1 + \|\mathbf{x}\|^2$ and a constant $\kappa = \tilde{\alpha} + \tilde{\beta}$, it yields that

$$P_{\boldsymbol{\theta} \in \mathcal{K}} V(\mathbf{x}) \leq \kappa V(\mathbf{x}). \tag{III}$$

Now we have verified the first condition (DRI1) by checking conditions (I),(II), and (III),

**(DRI2)** In what follows, we check the boundedness and Lipshitz conditions on the random-field function $\widetilde{H}(\boldsymbol{\theta}, \mathbf{x})$, where each subcomponent is defiend as $\widetilde{H}_i(\boldsymbol{\theta}, \mathbf{x}) = \theta(J_{\widetilde{U}}(\mathbf{x})) \left(1_{i=J_{\tilde{U}}(\mathbf{x})} - \theta(i)\right)$. Recall that $V = 1 + \|\mathbf{x}\|^2$, the compactness assumption A1 directly leads to

$$\sup_{\boldsymbol{\theta} \in \mathcal{K} \subset [0,1]^m} \|H(\boldsymbol{\theta}, \mathbf{x})\| \leq m V(\mathbf{x}). \tag{IV}$$

For any $\boldsymbol{\theta}_1, \boldsymbol{\theta}_2 \in \mathcal{K}$ and a fixed $\mathbf{x} \in \mathcal{X}$, it suffices for us to solely verify the $i$-th index, which is the index that maximizes $|\theta_1(i) - \theta_2(i)|$, then

$$
\begin{aligned}
|\widetilde{H}_i(\boldsymbol{\theta}_1, \mathbf{x}) - \widetilde{H}_i(\boldsymbol{\theta}_2, \mathbf{x})| &= \theta_1(J_{\widetilde{U}}(\mathbf{x})) \left(1_{i=J_{\tilde{U}}(\mathbf{x})} - \theta_1(i)\right) - \theta_2(J_{\widetilde{U}}(\mathbf{x})) \left(1_{i=J_{\tilde{U}}(\mathbf{x})} - \theta_2(i)\right) \\
&\leq |\theta_1(J_{\widetilde{U}}(\mathbf{x})) - \theta_2(J_{\widetilde{U}}(\mathbf{x}))| + |\theta_1(J_{\widetilde{U}}(\mathbf{x}))\theta_1(i) - \theta_2(J_{\widetilde{U}}(\mathbf{x}))\theta_2(i)| \\
&\leq \max_j \left(|\theta_1(j) - \theta_2(j)| + \theta_1(j)|\theta_1(i) - \theta_2(i)| + |\theta_1(j) - \theta_2(j)|\theta_2(i)\right) \\
&\leq 3|\theta_1(i) - \theta_2(i)|,
\end{aligned}
$$

where the last inequality holds since $\theta(i) \in (0, 1]$ for any $i \leq m$.

**(DRI3)** We proceed to verify the smoothness of the transitional kernel $P_{\boldsymbol{\theta}}(\mathbf{x}, \mathbf{y})$ with respect to $\boldsymbol{\theta}$. For any $\boldsymbol{\theta}_1, \boldsymbol{\theta}_2 \in \mathcal{K}$ and fixed $\mathbf{x}$ and $\mathbf{y}$, we have

$$
\begin{aligned}
&|P_{\boldsymbol{\theta}_1}(\mathbf{x}, \mathbf{y}) - P_{\boldsymbol{\theta}_2}(\mathbf{x}, \mathbf{y})| \\
&= \frac{1}{2\sqrt{(4\pi\epsilon)^{d/2}}} \mathbb{E}\left[e^{-\frac{\|\mathbf{y}-\mathbf{x}+\epsilon\nabla_{\mathbf{x}}\widetilde{L}(\mathbf{x},\boldsymbol{\theta}_1)\|^2}{4\epsilon}}|\mathbf{x}\right] - \frac{1}{2\sqrt{(4\pi\epsilon)^{d/2}}} \mathbb{E}\left[e^{-\frac{\|\mathbf{y}-\mathbf{x}+\epsilon\nabla_{\mathbf{x}}\widetilde{L}(\mathbf{x},\boldsymbol{\theta}_2)\|^2}{4\epsilon}}|\mathbf{x}\right] \\
&\lesssim |\|\mathbf{y} - \mathbf{x} + \epsilon\nabla_{\mathbf{x}}\widetilde{L}(\mathbf{x}, \boldsymbol{\theta}_1)\|^2 - \|\mathbf{y} - \mathbf{x} + \epsilon\nabla_{\mathbf{x}}\widetilde{L}(\mathbf{x}, \boldsymbol{\theta}_2)\|^2| \\
&\lesssim \|\nabla_{\mathbf{x}}\widetilde{L}(\mathbf{x}, \boldsymbol{\theta}_1) - \nabla_{\mathbf{x}}\widetilde{L}(\mathbf{x}, \boldsymbol{\theta}_2)\| \\
&\lesssim \|\boldsymbol{\theta}_1 - \boldsymbol{\theta}_2\|,
\end{aligned}
$$

where the first inequality (up to a finite constant) follows by $\|e^{\mathbf{x}} - e^{\mathbf{y}}\| \lesssim \|\mathbf{x} - \mathbf{y}\|$ for any $\mathbf{x}, \mathbf{y}$ in a compact space; the last inequality follows by the definition of the adaptive gradient in Eq.(15) and $\|\log(\mathbf{x}) - \log(\mathbf{y})\| \lesssim \|\mathbf{x} - \mathbf{y}\|$ by the compactness assumption A1.

For $f : \mathcal{X} \to \mathbb{R}^d$, define the norm $\|f\|_V = \sup_{\mathbf{x} \in \mathcal{X}} \frac{|f(\mathbf{x})|}{V(\mathbf{x})}$. Following the same technique proposed in Liang et al. (2007) (page 319), we can verify the last drift condition

$$\|P_{\boldsymbol{\theta}_1} f - P_{\boldsymbol{\theta}_2} f\|_V \leq C\|f\|_V \|\boldsymbol{\theta}_1 - \boldsymbol{\theta}_2\|, \ \ \forall f \in \mathcal{L}_V := \{f : \mathcal{X} \to \mathbb{R}^d, \|f\|_V < \infty\}. \tag{VI}$$

Having conditions (I), (II), $\cdots$ and (VI) verified, we are now able to prove the drift conditions proposed in Section 6 of Andrieu et al. (2005). ∎

Before we present the $L^2$ convergence of $\boldsymbol{\theta}_k$, we make some extra assumptions on the step size.

**Assumption A5 (Learning rate and step size)** *The learning rate $\{\epsilon_k\}_{k \in \mathbb{N}}$ is a positive non-increasing sequence of real numbers satisfying the conditions*

$$\lim_k \epsilon_k = 0, \quad \sum_{k=1}^{\infty} \epsilon_k = \infty.$$

*The step size $\{\omega_k\}_{k\in\mathbb{N}}$ is a positive non-increasing sequence of real numbers such that*

$$\lim_{k\to\infty}\omega_k = 0, \quad \sum_{k=1}^{\infty}\omega_k = +\infty, \quad \sum_{k=1}^{\infty}\omega_k^2 < +\infty. \tag{32}$$

*A practical strategy is to set $\omega_k := \mathcal{O}(k^{-\alpha})$ to satisfy the above conditions for any $\alpha \in (0.5, 1]$.*

The following is an application of Theorem 24 (page 246) (Benveniste et al., 1990) given stability conditions (Lemma 3).

**Lemma 7** ($L^2$ **convergence rate, restatement of Lemma 2**) *Assume Assumptions A1-A5 hold. For any $\boldsymbol{\theta}_0 \in \widetilde{\boldsymbol{\Theta}} \subset \boldsymbol{\Theta}$, a large $m$, small learning rates $\{\epsilon_k\}_{k=1}^{\infty}$, and step sizes $\{\omega_k\}_{k=1}^{\infty}$, $\{\boldsymbol{\theta}_k\}_{k=0}^{\infty}$ converges to $\widehat{\boldsymbol{\theta}}_\star$, where $\widehat{\boldsymbol{\theta}}_\star = \boldsymbol{\theta}_\star + \mathcal{O}\left(\sup_{\mathbf{x}}\mathrm{Var}(\xi_n(\mathbf{x})) + \sup_{k\geq k_0}\epsilon_k + \frac{1}{m}\right)$ for some $k_0$, such that*

$$\mathbb{E}\left[\|\boldsymbol{\theta}_k - \widehat{\boldsymbol{\theta}}_\star\|^2\right] = \mathcal{O}\left(\omega_k\right).$$

The theoretical novelty is that we treat the biased $\widehat{\boldsymbol{\theta}}_\star$ as the equilibrium of the continuous system instead of analyzing how far we are away from $\boldsymbol{\theta}_\star$ in all aspects as in Theorem 1 (Deng et al., 2020b). This enables us to directly apply Theorem 24 (page 246). Nevertheless, it can be interpreted as a special case of Theorem 1 (Deng et al., 2020b) except that there are no perturbation terms and the equilibrium is $\widehat{\boldsymbol{\theta}}_\star$ instead of $\boldsymbol{\theta}_\star$.

## C  GAUSSIAN APPROXIMATION

### C.1  PRELIMINARY: SUFFICIENT CONDITIONS FOR WEAK CONVERGENCE

To formally prove the asymptotic normality of the stochastic approximation process $\omega_k^{-1/2}(\boldsymbol{\theta}_k - \widehat{\boldsymbol{\theta}}_\star)$, we first lay out a preliminary result (Theorem 1 of Pelletier (1998)) that provides sufficient conditions to guarantee the weak convergence.

**Lemma 8 (Sufficient Conditions)** *Consider a stochastic algorithm as follows*

$$\boldsymbol{\theta}_{k+1} = \boldsymbol{\theta}_k + \omega_{k+1}h(\boldsymbol{\theta}_k) + \omega_{k+1}\widetilde{\boldsymbol{\nu}}_{k+1} + \omega_{k+1}\boldsymbol{e}_{k+1},$$

*where $\widetilde{\boldsymbol{\nu}}_{k+1}$ denotes a perturbation and $\boldsymbol{e}_{k+1}$ is a random noise. Given three conditions (**C1**), (**C2**), and (**C3**) defined below, we have the desired weak convergence result*

$$\omega^{-\frac{1}{2}}(\boldsymbol{\theta}_k - \widehat{\boldsymbol{\theta}}_\star) \Rightarrow \mathcal{N}(0, \boldsymbol{\Sigma}), \tag{33}$$

*where $\boldsymbol{\Sigma} = \int_0^{\infty} e^{th_{\boldsymbol{\theta}_\star}} \circ \boldsymbol{R} \circ e^{th_{\boldsymbol{\theta}_\star}^\top} dt$, $\boldsymbol{R}$ denotes the limiting covariance of the martingale $\lim_{k\to\infty}\mathbb{E}[\boldsymbol{e}_{k+1}\boldsymbol{e}_{k+1}^\top|\mathcal{F}_k]$ and $\mathcal{F}_k$ is the $\sigma$-algebra of the events up to iteration $k$, $h_{\boldsymbol{\theta}_\star} = h_{\boldsymbol{\theta}}(\widehat{\boldsymbol{\theta}}_\star)+\widehat{\xi}\boldsymbol{I}$, $\widehat{\xi} = \lim_{k\to\infty}\frac{\omega_k^{0.5}-\omega_{k+1}^{0.5}}{\omega_k^{1.5}}$.[†]*

*(**C1**) There exists an equilibrium point $\widehat{\boldsymbol{\theta}}_\star$ and a stable matrix $h_{\boldsymbol{\theta}_\star} := h_{\boldsymbol{\theta}}(\widehat{\boldsymbol{\theta}}_\star) \in \mathbb{R}^{m\times m}$ such that for any $\boldsymbol{\theta} \in \{\boldsymbol{\theta} : \|\boldsymbol{\theta} - \widehat{\boldsymbol{\theta}}_\star\| \leq \widetilde{M}\}$ for some $\widetilde{M} > 0$, the mean-field function $h : \mathbb{R}^m \to \mathbb{R}^m$ satisfies*

$$h(\widehat{\boldsymbol{\theta}}_\star) = 0$$
$$\|h(\boldsymbol{\theta}) - h_{\boldsymbol{\theta}_\star}(\boldsymbol{\theta} - \widehat{\boldsymbol{\theta}}_\star)\| \lesssim \|\boldsymbol{\theta} - \widehat{\boldsymbol{\theta}}_\star\|^2,$$

*(**C2**) The step size $\omega_k$ decays with an order $\alpha \in (0, 1]$ such that $\omega_k = \mathcal{O}(k^{-\alpha})$.*

*(**C3**) Assumptions on the disturbances . There exists constants $\widetilde{M} > 0$ and $\widetilde{\alpha} > 2$ such that*

$$\mathbb{E}\left[\boldsymbol{e}_{k+1}|\mathcal{F}_k\right]\mathbf{1}_{\{\|\boldsymbol{\theta}-\widehat{\boldsymbol{\theta}}_\star\|\leq\widetilde{M}\}} = 0, \tag{$I_1$}$$

$$\sup_k\mathbb{E}\left[\|\boldsymbol{e}_{k+1}\|^{\widetilde{\alpha}}|\mathcal{F}_k\right]\mathbf{1}_{\{\|\boldsymbol{\theta}-\widehat{\boldsymbol{\theta}}_\star\|\leq\widetilde{M}\}} < \infty, \tag{$I_2$}$$

$$\mathbb{E}\left[\omega_k^{-1}\|\widetilde{\boldsymbol{\nu}}_{k+1}\|^2\right]\mathbf{1}_{\{\|\boldsymbol{\theta}-\widehat{\boldsymbol{\theta}}_\star\|\leq\widetilde{M}\}} \to 0, \tag{$II$}$$

$$\mathbb{E}\left[\boldsymbol{e}_{k+1}\boldsymbol{e}_{k+1}^\top|\mathcal{F}_k\right]\mathbf{1}_{\{\|\boldsymbol{\theta}-\widehat{\boldsymbol{\theta}}_\star\|\leq\widetilde{M}\}} \to \boldsymbol{R}. \tag{$III$}$$

---

[†]For example, $\widehat{\xi} = 0$ if $\omega_k = \mathcal{O}(k^{-\alpha})$, where $\alpha \in (0.5, 1]$ and $\widehat{\xi} = \frac{k_0}{2}$ if $\omega_k = \frac{k_0}{k}$.

**Remark 2** *By the definition of the mean-field function $h(\boldsymbol{\theta})$ in Eq.(26), it is easy to verify the condition C1. Moreover, Assumption A5 also fulfills the condition C2. Then, the proof hinges on the verification of the condition C3.*

## C.2 PRELIMINARY: CONVERGENCE OF THE COVARIANCE ESTIMATORS

In particular, to verify the condition $\mathbb{E}\left[e_{k+1}e_{k+1}^{\top}|\mathcal{F}_k\right]\mathbf{1}_{\{\|\boldsymbol{\theta}-\widehat{\boldsymbol{\theta}}_\star\|\leq\widetilde{M}\}}\rightarrow\boldsymbol{R}$, , we study the convergence of the empirical sample mean $\mathbb{E}[f(\mathbf{x}_k)]$ for a test function $f$ to the posterior expectation $\bar{f}=\int_\mathcal{X}f(\mathbf{x})\varpi_{\widehat{\boldsymbol{\theta}}_\star}(\mathbf{x})(d\mathbf{x})$. Poisson's equation is often used to characterize the fluctuation between $f(\mathbf{x})$ and $\bar{f}$:

$$\mathcal{L}g(\mathbf{x})=f(\mathbf{x})-\bar{f},\tag{34}$$

where $\mathcal{L}$ refers to an infinitesimal generator and $g(\mathbf{x})$ denotes the solution of the Poisson's equation. Similar to the proof of Lemma 6, the existence of the solution of the Poisson's equation has been established in (Mattingly et al., 2002; Vollmer et al., 2016). Moreover, the perturbations of $\mathbb{E}[f(\mathbf{x}_k)]-\bar{f}$ are properly bounded given regularity properties for $g(\mathbf{x})$, where the 0-th, 1st, and 2nd order of the regularity properties has been established in Erdogdu et al. (2018).

The following result helps us to identify the convergence of the covariance estimators, which is adapted from Theorem 5 (Chen et al., 2015) with decreasing learning rates $\{\epsilon_k\}_{k\geq1}$. The gradient biases from Theorem 2 (Chen et al., 2015) are also included to handle the adaptive biases.

**Lemma 9 (Convergence of the Covariance Estimators)** *Suppose Assumptions A1-A5 hold. For any $\boldsymbol{\theta}_0\in\widetilde{\boldsymbol{\Theta}}\subset\boldsymbol{\Theta}$, a large $m$, small learning rates $\{\epsilon_k\}_{k=1}^\infty$, step sizes $\{\omega_k\}_{k=1}^\infty$ and any bounded function $f$, we have*

$$\left|\mathbb{E}\left[f(\mathbf{x}_k)\right]-\int_\mathcal{X}f(\mathbf{x})\varpi_{\widehat{\boldsymbol{\theta}}_\star}(\mathbf{x})d\mathbf{x}\right|\rightarrow0,$$

*where $\varpi_{\widehat{\boldsymbol{\theta}}_\star}(\mathbf{x})$ is the invariant measure simulated via SGLD that approximates $\varpi_{\widetilde{\Psi}_{\boldsymbol{\theta}_\star}}(\mathbf{x})\propto\frac{\pi(\mathbf{x})}{\theta_\star^\zeta(J(\mathbf{x}))}$.*

**Proof** We study the single-chain CSGLD and reformulate the adaptive algorithm as follows:

$$\begin{aligned}\mathbf{x}_{k+1}&=\mathbf{x}_k-\epsilon_k\nabla_\mathbf{x}\widetilde{L}(\mathbf{x}_k,\boldsymbol{\theta}_k)+\mathcal{N}(0,2\epsilon_k\tau\boldsymbol{I})\\&=\mathbf{x}_k-\epsilon_k\left(\nabla_\mathbf{x}\widetilde{L}(\mathbf{x}_k,\widehat{\boldsymbol{\theta}}_\star)+\Upsilon(\mathbf{x}_k,\boldsymbol{\theta}_k)\right)+\mathcal{N}(0,2\epsilon_k\tau\boldsymbol{I}),\end{aligned}$$

where $\nabla_\mathbf{x}\widetilde{L}(\mathbf{x},\boldsymbol{\theta})=\frac{N}{n}\left[1+\frac{\zeta\tau}{\Delta u}\left(\log\theta(J(\mathbf{x}))-\log\theta((J(\mathbf{x})-1)\vee1)\right)\right]\nabla_\mathbf{x}\widetilde{U}(\mathbf{x})$ [‡], $\nabla_\mathbf{x}\widetilde{L}(\mathbf{x},\boldsymbol{\theta})$ is defined in Section B.1 and the bias term is given by $\Upsilon(\mathbf{x}_k,\boldsymbol{\theta}_k)=\nabla_\mathbf{x}\widetilde{L}(\mathbf{x}_k,\boldsymbol{\theta}_k)-\nabla_\mathbf{x}\widetilde{L}(\mathbf{x}_k,\widehat{\boldsymbol{\theta}}_\star)$.

Then, by Jensen's inequality and Lemma 7, we have

$$\begin{aligned}\|\mathbb{E}[\Upsilon(\mathbf{x}_k,\boldsymbol{\theta}_k)]\|&\leq\mathbb{E}[\|\nabla_\mathbf{x}\widetilde{L}(\mathbf{x}_k,\boldsymbol{\theta}_k)-\nabla_\mathbf{x}\widetilde{L}(\mathbf{x}_k,\widehat{\boldsymbol{\theta}}_\star)\|]\\&\lesssim\mathbb{E}[\|\boldsymbol{\theta}_k-\widehat{\boldsymbol{\theta}}_\star\|]\leq\sqrt{\mathbb{E}[\|\boldsymbol{\theta}_k-\widehat{\boldsymbol{\theta}}_\star\|^2]}\leq\mathcal{O}\left(\sqrt{\omega_k}\right).\end{aligned}\tag{35}$$

Combining Eq.(35) and Theorem 5 (Chen et al., 2015), we have

$$\begin{aligned}\left|\mathbb{E}\left[f(\mathbf{x}_k)\right]-\int_\mathcal{X}f(\mathbf{x})\varpi_{\widehat{\boldsymbol{\theta}}_\star}(\mathbf{x})d\mathbf{x}\right|&=\mathcal{O}\left(\frac{1}{\sum_i^k\epsilon_i}+\frac{\sum_{i=1}^k\omega_i\|\mathbb{E}[\Upsilon(\mathbf{x}_i,\boldsymbol{\theta}_i)]\|}{\sum_i^k\omega_i}+\frac{\sum_i^k\epsilon_i^2}{\sum_i^k\epsilon_i}\right)\\&\rightarrow0,\text{ as }k\rightarrow\infty,\end{aligned}$$

where the last argument directly follows from the conditions on learning rates and step sizes in Assumption A5. ∎

---

[‡] $J(\mathbf{x})=\sum_{i=1}^m i\mathbf{1}_{u_{i-1}<U(\mathbf{x})\leq u_i}$, where the exact energy function $U(\mathbf{x})$ is selected.

### C.3 PROOF OF THEOREM 1

Recall that the stochastic approximation based on a single process follows from

$$
\begin{aligned}
&\boldsymbol{\theta}_{k+1}\\
&= \boldsymbol{\theta}_k + \omega_{k+1} H(\boldsymbol{\theta}_k, \mathbf{x}_{k+1})\\
&= \boldsymbol{\theta}_k + \omega_{k+1} h(\boldsymbol{\theta}_k) + \omega_{k+1}\left(\mu_{\boldsymbol{\theta}_k}(\mathbf{x}_{k+1}) - \Pi_{\boldsymbol{\theta}_k}\mu_{\boldsymbol{\theta}_k}(\mathbf{x}_{k+1})\right)\\
&= \boldsymbol{\theta}_k + \omega_{k+1} h(\boldsymbol{\theta}_k)\\
&\quad + \omega_{k+1}\underbrace{\left(\Pi_{\boldsymbol{\theta}_{k+1}}\mu_{\boldsymbol{\theta}_{k+1}}(\mathbf{x}_{k+1}) - \Pi_{\boldsymbol{\theta}_k}\mu_{\boldsymbol{\theta}_k}(\mathbf{x}_{k+1}) + \frac{\omega_{k+2}-\omega_{k+1}}{\omega_{k+1}}\Pi_{\boldsymbol{\theta}_{k+1}}\mu_{\boldsymbol{\theta}_{k+1}}(\mathbf{x}_{k+1})\right)}_{\boldsymbol{\nu}_{k+1}}\\
&\quad + \omega_{k+1}\left(\underbrace{\frac{1}{\omega_{k+1}}\left(\omega_{k+1}\Pi_{\boldsymbol{\theta}_k}\mu_{\boldsymbol{\theta}_k}(\mathbf{x}_k) - \omega_{k+2}\Pi_{\boldsymbol{\theta}_{k+1}}\mu_{\boldsymbol{\theta}_{k+1}}(\mathbf{x}_{k+1})\right)}_{\varsigma_{k+1}} + \underbrace{\mu_{\boldsymbol{\theta}_k}(\mathbf{x}_{k+1}) - \Pi_{\boldsymbol{\theta}_k}\mu_{\boldsymbol{\theta}_k}(\mathbf{x}_k)}_{\boldsymbol{e}_{k+1}}\right)\\
&= \boldsymbol{\theta}_k + \omega_{k+1} h(\boldsymbol{\theta}_k) + \omega_{k+1}\underbrace{(\boldsymbol{\nu}_{k+1} + \varsigma_{k+1})}_{\text{perturbation}} + \omega_{k+1}\underbrace{\boldsymbol{e}_{k+1}}_{\text{martingale}},
\end{aligned}
\tag{36}
$$

where the second equality holds from the solution of Poisson's equation in Eq.(30).

We denote $\ddot{\boldsymbol{\theta}}_k = \boldsymbol{\theta}_k + \omega_{k+1}\Pi_{\boldsymbol{\theta}_k}\mu_{\boldsymbol{\theta}_k}(\mathbf{x}_k)$. Adding $\omega_{k+2}\Pi_{\boldsymbol{\theta}_{k+1}}\mu_{\boldsymbol{\theta}_{k+1}}(\mathbf{x}_{k+1})$ on both sides of Eq.(36), we have

$$
\begin{aligned}
&\ddot{\boldsymbol{\theta}}_{k+1}\\
&= \ddot{\boldsymbol{\theta}}_k + \omega_{k+1} h(\boldsymbol{\theta}_k) + \omega_{k+1}\left(\boldsymbol{\nu}_{k+1} + \boldsymbol{e}_{k+1} + \varsigma_{k+1}\right) + \omega_{k+2}\Pi_{\boldsymbol{\theta}_{k+1}}\mu_{\boldsymbol{\theta}_{k+1}}(\mathbf{x}_{k+1}) - \omega_{k+1}\Pi_{\boldsymbol{\theta}_k}\mu_{\boldsymbol{\theta}_k}(\mathbf{x}_k)\\
&= \ddot{\boldsymbol{\theta}}_k + \omega_{k+1} h(\boldsymbol{\theta}_k) + \omega_{k+1}\left(\boldsymbol{\nu}_{k+1} + \boldsymbol{e}_{k+1}\right)\\
&= \ddot{\boldsymbol{\theta}}_k + \omega_{k+1} h(\ddot{\boldsymbol{\theta}}_k) + \omega_{k+1}\left(\tilde{\boldsymbol{\nu}}_{k+1} + \boldsymbol{e}_{k+1}\right),
\end{aligned}
\tag{37}
$$

where $\tilde{\boldsymbol{\nu}}_{k+1} = \boldsymbol{\nu}_{k+1} + h(\boldsymbol{\theta}_k) - h(\ddot{\boldsymbol{\theta}}_k)$. Next, we proceed to verify the conditions in **C3**.

(I) By the martingale difference property of $\{\boldsymbol{e}_k\}$ and the compactness assumption A1, we know that for any $\widetilde{\alpha} > 2$

$$
\mathbb{E}[\boldsymbol{e}_{k+1}|\mathcal{F}_k] = \mathbf{0}, \quad \sup_{k\geq 0}\mathbb{E}[\|\boldsymbol{e}_{k+1}\|^{\widetilde{\alpha}}|\mathcal{F}_k] < \infty.
\tag{I}
$$

(II) By the definition of $h(\boldsymbol{\theta}_k)$ in Eq.(26), we can easily check that $h(\boldsymbol{\theta}_k)$ is Lipschitz continuous in a neighborhood of $\widehat{\boldsymbol{\theta}}_\star$. Combining Eq.(31), we have $\|h(\boldsymbol{\theta}_k) - h(\ddot{\boldsymbol{\theta}}_k)\| = \mathcal{O}(\|\boldsymbol{\theta}_k - \ddot{\boldsymbol{\theta}}_k\|) = \mathcal{O}(\|\omega_{k+1}\Pi_{\boldsymbol{\theta}_k}\mu_{\boldsymbol{\theta}_k}(\mathbf{x}_k)\|) = \mathcal{O}(\omega_{k+1})$. Then $\mathbb{E}[\|\boldsymbol{\nu}_{k+1}\|] \leq C\|\boldsymbol{\theta}_k - \ddot{\boldsymbol{\theta}}_k\| + \mathcal{O}(\omega_{k+2}) = \mathcal{O}(\omega_{k+1})$ by the step size condition Eq.(32). In what follows, we can verify

$$
\mathbb{E}\left[\frac{\|\tilde{\boldsymbol{\nu}}_{k+1}\|^2}{\omega_k}\right] \leq 2\mathbb{E}\left[\frac{\|\boldsymbol{\nu}_{k+1}\|^2}{\omega_k}\right] + 2\mathbb{E}\left[\frac{\|h(\boldsymbol{\theta}_k) - h(\ddot{\boldsymbol{\theta}}_k)\|^2}{\omega_k}\right] = \mathcal{O}(\omega_k) \to 0.
\tag{II}
$$

(III) For the martingale difference noise $\boldsymbol{e}_{k+1} = \mu_{\boldsymbol{\theta}_k}(\mathbf{x}_{k+1}) - \Pi_{\boldsymbol{\theta}_k}\mu_{\boldsymbol{\theta}_k}(\mathbf{x}_k)$ with mean 0, we have

$$
\mathbb{E}[\boldsymbol{e}_{k+1}\boldsymbol{e}_{k+1}^\top|\mathcal{F}_k] = \mathbb{E}[\mu_{\boldsymbol{\theta}_k}(\mathbf{x}_{k+1})\mu_{\boldsymbol{\theta}_k}(\mathbf{x}_{k+1})^\top|\mathcal{F}_k] - \Pi_{\boldsymbol{\theta}_k}\mu_{\boldsymbol{\theta}_k}(\mathbf{x}_k)\Pi_{\boldsymbol{\theta}_k}\mu_{\boldsymbol{\theta}_k}(\mathbf{x}_k)^\top.
$$

We denote $\mathbb{E}[\boldsymbol{e}_{k+1}\boldsymbol{e}_{k+1}^\top|\mathcal{F}_k]$ by a function $f(\mathbf{x}_k)$. Applying Lemma 9, we have

$$
\mathbb{E}[\boldsymbol{e}_{k+1}\boldsymbol{e}_{k+1}^\top|\mathcal{F}_k] = \mathbb{E}[f(\mathbf{x}_k)] \to \int f(\mathbf{x})\varpi_{\widehat{\boldsymbol{\theta}}_\star}d\mathbf{x} = \lim_{k\to\infty}\mathbb{E}[\boldsymbol{e}_{k+1}\boldsymbol{e}_{k+1}^\top|\mathcal{F}_k] := \boldsymbol{R},
\tag{III}
$$

where $\boldsymbol{R} := \boldsymbol{R}(\widehat{\boldsymbol{\theta}}_\star)$ and $\boldsymbol{R}(\boldsymbol{\theta})$ is also equivalent to $\sum_{k=-\infty}^{\infty}\text{Cov}_{\boldsymbol{\theta}}(H(\boldsymbol{\theta}, \mathbf{x}_k), H(\boldsymbol{\theta}, \mathbf{x}_0))$.

Having the conditions C1, C2 and C3 verified, we apply Lemma 8 and have the following weak convergence for $\ddot{\boldsymbol{\theta}}_k$

$$
\omega_k^{-1/2}(\ddot{\boldsymbol{\theta}}_k - \widehat{\boldsymbol{\theta}}_\star) \Rightarrow \mathcal{N}(0, \boldsymbol{\Sigma}),
$$

where $\boldsymbol{\Sigma} = \int_0^\infty e^{th_{\boldsymbol{\theta}_\star}} \circ \boldsymbol{R} \circ e^{th_{\boldsymbol{\theta}_\star}^\top} dt$ and $h_{\boldsymbol{\theta}_\star} = h_{\boldsymbol{\theta}}(\widehat{\boldsymbol{\theta}}_\star) + \widehat{\xi} \boldsymbol{I}, \widehat{\xi} = \lim_{k\to\infty} \frac{\omega_k^{0.5} - \omega_{k+1}^{0.5}}{\omega_k^{1.5}}$.

Considering the definition that $\ddot{\boldsymbol{\theta}}_k = \boldsymbol{\theta}_k + \omega_{k+1} \Pi_{\boldsymbol{\theta}_k} \mu_{\boldsymbol{\theta}_k}(\mathbf{x}_k)$ and $\mathbb{E}[\|\Pi_{\boldsymbol{\theta}_k} \mu_{\boldsymbol{\theta}_k}(\mathbf{x}_k)\|]$ is uniformly bounded by Eq.(31), we have

$$\omega_k^{1/2} \Pi_{\boldsymbol{\theta}_k} \mu_{\boldsymbol{\theta}_k}(\mathbf{x}_k) \to 0 \quad \text{in probability.}$$

By Slutsky's theorem, we eventually have the desired result

$$\omega_k^{-1/2}(\boldsymbol{\theta}_k - \widehat{\boldsymbol{\theta}}_\star) \Rightarrow \mathcal{N}(0, \boldsymbol{\Sigma}).$$

where the step size $\omega_k$ decays with an order $\alpha \in (0.5, 1]$ such that $\omega_k = \mathcal{O}(k^{-\alpha})$. ∎

# D  MORE ON EXPERIMENTS

## D.1  MODE EXPLORATION ON MNIST VIA THE SCALABLE RANDOM-FIELD FUNCTION

For the network structure, we follow Jarrett et al. (2009) and choose a standard convolutional neural network (CNN). Such a CNN has two convolutional (conv) layers and two fully-connected (FC) layers. The two conv layers has 32 and 64 feature maps, respectively. The FC layers both have 50 hidden nodes and the network has 5 outputs. A large batch size of 2500 is selected to reduce the gradient noise and reduce the stochastic approximation bias. We fix $\zeta = 3e4$ and weight decay 25. For simplicity, we choose 100,000 partitions and $\Delta u = 10$. The step size follows $\omega_k = \min\{0.01, \frac{1}{k^{0.6}+100}\}$.

## D.2  SIMULATIONS OF MULTI-MODAL DISTRIBUTIONS

The target density function is given by $\pi(\mathbf{x}) \propto \exp(-U(\mathbf{x}))$, where $\mathbf{x} = (x_1, x_2)$ and $U(\mathbf{x})$ follows $U(\mathbf{x}) = 0.2(x_1^2 + x_2^2) - 2(\cos(2\pi x_1) + \cos(2\pi x_2))$. We also include a regularization term $L(x) = \mathbb{I}_{(x_1^2+x_2^2)>20} \times ((x_1^2 + x_2^2) - 20)$. This design leads to a highly multi-modal distribution with 25 isolated modes. Figure 5 shows the contour and the 3-D plot of the target density. The ICSGLD and baseline algorithms are applied to this example. For IC-SGLD, we set $\epsilon_k = 3e^{-3}$, $\tau = 1$, $\zeta = 0.75$ and total number of iterations$= 8e^4$. Besides, we partition the sample space into 100 subregions with bandwidth $\Delta u = 0.125$ and set $\omega_k = \min(3e^{-3}, \frac{1}{k^{0.6}+100})$.

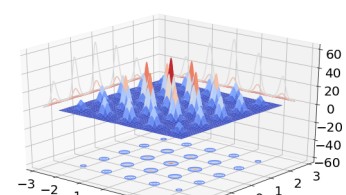

Figure 5: Target density.

For comparison, we run the baseline algorithms under similar settings. For CSGLD, we run a single process 5 times of the time budget and all the settings are the same as those used by ICSGLD. For reSGLD, we run five parallel chains with learning rates $0.001, 0.002, \cdots, 0.005$ and temperatures $1, 2, \cdots, 5$, respectively. We estimate the correction every 100 iterations. We fix the initial correction 30 and choose the same step size for the stochastic approximation as in ICSGLD. For SGLD, we run five chains in parallel with the learning rate $3e^{-3}$ and a temperature of 1. For cycSGLD, we run a single-chain with 5 times of the time budget. We set the initial learning rate as $1e^{-2}$ and choose 10 cycles. For the particle-based SVGD, we run five chains in parallel. For each chain, we initialize 100 particles as being drawn from a uniform distribution over a rectangle. The learning rate is set to $3e^{-3}$.

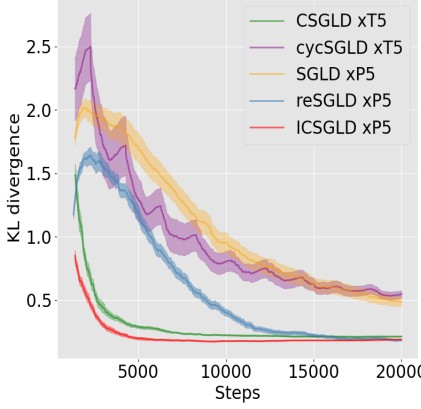 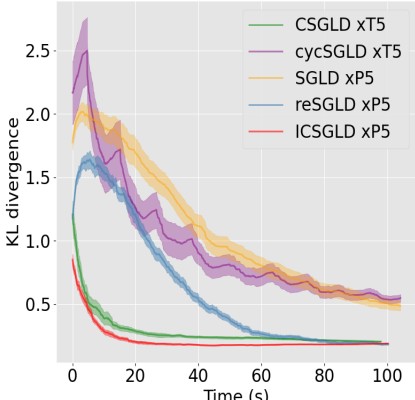

Figure 6: Estimation KL divergence versus time steps for ICSGLD and baseline methods. We repeat experiments 20 times.

To compare the convergence rates in terms of *running steps* and *time* between ICSGLD and other algorithms, we repeat each algorithm 20 times and calculate the mean and standard error over 20 trials. Note that we run all the algorithms based on 5 parallel chains ($\times$P5) except that cycSGLD and CSGLD are run in a single-chain with 5 times of time budget ($\times$T5) and the steps and running

time are also scaled accordingly. Figure 6 shows that the vanilla SGLD×P5 converges the slowest among the five algorithms due to the lack of mechanism to escape local traps; cycSGLD×T5 slightly alleviates that problem by adopting cyclical learning rates; reSGLD×P5 greatly accelerates the computations by utilizing high-temperature chains for exploration and low-temperature chains for exploitation, but the large correction term inevitably slows down the convergence; ICSGLD×P5 converges faster than all the others and the noisy energy estimators only induce a bias for the latent variables and don't affect the convergence rate significantly.

For the particle-based SVGD method, since more particles require expensive computations while fewer particles lead to a crude approximation. Therefore, we don't show the convergence of SVGD and only compare the Monte Carlo methods.

### D.3 Deep contextual bandits on mushroom tasks

For the UCI Mushroom data set, each mushroom is either edible or poisonous. Eating an edible mushroom yields a reward of 5, but eating a poisonous mushroom has a 50% chance to result in a reward of -35 and a reward of 5 otherwise. Eating nothing results in 0 reward. All the agents use the same architecture. In particular, we fit a two-layer neural network with 100 neurons each and ReLU activation functions. The input of the network is a feature vector with dimension 22 (context) and there are 2 outputs, representing the predicted reward for eating or not eating a mushroom. The mean squared loss is adopted for training the models. We initialize 1024 data points and keep a data buffer of size 4096 as the training proceeds. The size of the mini-batch data is set to 512. To adapt to online scenarios, we train models after every 20 new observations.

We choose one $\epsilon$-greedy policy (EpsGreedy) based on the RMSProp optimizer with a decaying learning rate (Riquelme et al., 2018) as a baseline. Two variational methods, namely stochastic gradient descent with a constant learning rate (ConstSGD) (Mandt et al., 2017) and Monte Carlo Dropout (Dropout) (Gal & Ghahramani, 2016) are compared to approximate the posterior distribution. For the sampling algorithms, we include preconditioned SGLD (pSGLD) (Li et al., 2016), preconditioned CSGLD (pCSGLD) (Deng et al., 2020b), and preconditioned ICSGLD (pICSGLD). Note that all the algorithms run 4 parallel chains with average outputs (×P4) except that pCSGLD runs a single-chain with 4 times of computational budget (×T4). In particular for the two contour algorithms, we set $\zeta = 20$ and choose a constant step size for the stochastic approximation to fit for the time-varying posterior distributions. For more details on the experimental setups, we refer readers to section D in the supplementary material.

We report the experimental setups for each algorithm. Similar to Table 2 of Riquelme et al. (2018), the inclusion of advanced techniques may change the optimal settings of the hyperparameters. Nevertheless, we try to report the best setups for each individual algorithm. We train each algorithm 2000 steps. We initialize 1024 mushrooms and keep a data buffer of size 4096 as the training proceeds. For each step, we are given 20 random mushrooms and train the model 16 iterations every step for the parallel algorithms (×P4); we train pCSGLD×T4 64 iterations every step.

EpsGreedy decays the learning rate by a factor of 0.999 every step; by contrast, all the others choose a fixed learning rate. RMSprop adopts a regularizer of 0.001 and a learning rate of 0.01 to learn the preconditioners. Dropout proposes a 50% dropout rate and each subprocess simulates 5 models for predictions. For the two importance sampling (IS) algorithms, we partition the energy space into $m = 100$ subregions and set the energy depth $\Delta u$ as 10. We fix the hyperrameter $\zeta = 20$. The step sizes for pICSGLD×P4 and pCSGLD×T4 are chosen as 0.03 and 0.006, respectively. A proper regularizer is adopted for the low importance weights. See Table 2 for details.

Table 2: Details of the experimental setups.

| Algorithm | Learning rate | Temperature | RMSprop | IS | Train | Dropout | $\epsilon$-Greedy |
|---|---|---|---|---|---|---|---|
| EpsGreedy×P4 | 5e-7 (0.999) | 0 | Yes | No | 16 | No | 0.3% |
| ConstSGD×P4 | 1e-6 | 0 | No | No | 16 | No | No |
| Dropout×P4 | 1e-6 | 0 | No | No | 16 | Yes (50%) | No |
| pCSGLD×T4 | 5e-8 | 0.3 | Yes | Yes | 64 | No | No |
| pSGLD×P4 | 3e-7 | 0.3 | Yes | No | 16 | No | No |
| pICSGLD×P4 | 3e-7 | 0.3 | Yes | Yes | 16 | No | No |

### D.4 UNCERTAINTY ESTIMATION

All the algorithms, excluding M-SGD×P4, choose a temperature of 0.0003 [†]. We run the parallel algorithms 500 epochs (×P4) and run the single-chain algorithms 2000 epochs (×T4). The initial learning rate is 2e-6 (Bayesian settings), which corresponds to the standard 0.1 for averaged data likelihood.

We train cycSGHMC×T4 and MultiSWAG×T4 based on the cosine learning rates with 10 cycles. The learning rate in the last 15% of each cycle is fixed at a constant value. MultiSWAG simulates 10 random models at the end of each cycle. M-SGD×P4 follows the same cosine learning rate strategy with one cycle.

reSGHMC×P4 proposes swaps between neighboring chains and requires a fixed correction of 4000 for ResNet20, 32, and 56 and a correction of 1000 for WRN-16-8. The learning rate is annealed at 250 and 375 epochs with a factor of 0.2. ICSGHMC×P4 also applies the same learning rate. We choose $m = 200$ and $\Delta u = 200$ for ResNet20, 32, and 56 and $\Delta u = 60$ for WRN-16-8. Proper regularizations may be applied to the importance weights and gradient multipliers for training deep neural networks.

Variance reduction (Deng et al., 2021a) only applies to reSGHMC×P4 and ICSGHMC×P4 because they are the only two algorithms that require accurate estimations of the energy. We only update control variates every 2 epochs in the last 100 epochs, which maintain a reasonable training time and a higher reduction of variance due to a small learning rate. Other algorithms yield a worse performance when variance reduction is applied to the gradients.

### D.5 EMPIRICAL VALIDATION OF REDUCED VARIANCE

To compare the $\theta$'s learned from ICSGLD and CSGLD, we try to simulate from a Gaussian mixture distribution $0.4N(-6, 1) + 0.6N(4, 1)$, where $N(u, v)$ denotes a Gaussian distribution with mean $u$ and standard deviation $v$. We fix $\zeta = 0.9$ and $\Delta u = 1$. We run ICSGLD with 1,000,000 iterations based on 10 interacting parallel chains and run CSGLD with 10,000,000 iterations using a single chain. We refer to them as ICSGLD×P10 and CSGLD×T10, respectively. The rest of the settings follows from the experimental setup in section 4.1 (Deng et al., 2020a).

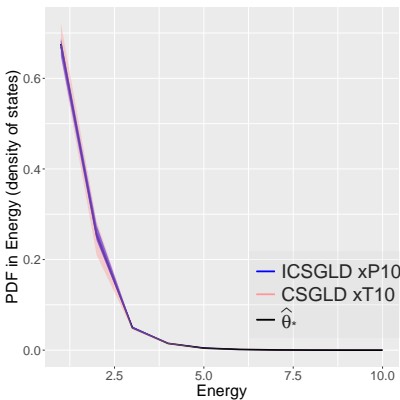

To measure the variance of the estimates, we repeated the experiments 10 times and present the mean and two standard deviations for both CSGLD×T10 and ICSGLD×P10 in Figure 7. The results indicate that both estimates of $\theta^\zeta$ (by CSGLD and ICSGLD) converge to the equilibrium that approximates the ground truth of the density of states. Notably, ICSGLD×P10 yields a *significantly smaller variance* than CSGLD×T10, but with the same computational budget. This shows the clear advantage of ICSGLD (many interacting short runs) over CSGLD (a single long run) in tackling the *large variance issue* for importance sampling.

Figure 7: ICSGLD v.s. CSGLD.

---

[†]We use various data augmentation techniques, such as random flipping, cropping, and random erasing (Zhong et al., 2017). This leads to a much more concentrated posterior and requires a very low temperature.

