# OpenReview forum: "Interacting Contour Stochastic Gradient Langevin Dynamics"
_ICLR.cc/2022/Conference — ICLR 2022 Poster_

### Official Review · Reviewer_9d9U · 2021-11-02

**Correctness:** 4
**Technical Novelty And Significance:** 3
**Empirical Novelty And Significance:** 3
**Recommendation:** 6
**Confidence:** 3

**Main Review:**

The strength of the paper is that it proposes a new algorithm interacting contour stochastic gradient Langevin dynamics that can be shown theoretically more efficient than contour stochastic gradient Langevin dynamics. The paper is well written, and it has both theoretical justification and numerical support.

The weakness of the paper is that many results seem to be adapted from Deng et al. (2020) for the contour stochastic gradient Langevin dynamics. Even though, a new algorithm is proposed, it should be considered as an extension to the previous work Deng et al. (2020), and some similar theoretical estimates can be found in Deng et al. (2020) and the technical novelty is not clear.

**Summary Of The Paper:**

The paper proposed an interacting contour stochastic gradient Langevin dynamics that extends the contour stochastic gradient Langevin dynamics with efficient interactions in Deng et al. (2020). They obtain asymptotic normality as well as show that this proposed algorithm can be more efficient than  the contour stochastic gradient Langevin dynamics with efficient interactions. The theoretical results are complemented by the numerical experiments.

**Summary Of The Review:**

(1) Page 2. For ICSGLD and CSGLD, I have one naive question. You are sampling (2) with a flattened density compared to the original Gibbs distribution $\pi$. If you are sampling a modified distribution instead, do you need to show that the new target distribution is close to the original distribution?

(2) Page 2. I am a bit confused with the notation here. In equation (2), you have $\Psi^{\zeta}_{\theta}(U(x))$ and then in the bottom of page 2, you have $\tilde{U}_{\Psi_{\theta_{k}}}$. What's the relation between these two different notations?

(3) Page 3. In your Algorithm 1, there is a stochastic approximation step for $\theta_{k}$. In your Assumption A2 in the appendix, you assume that the space $\Theta$ is compact. How do you guarantee your $\theta_{k}$ lives in a compact space since you are doing $k\rightarrow\infty$ analysis.

(4) Your Theorem 1 and Corollary 1 for the asymptotic normality is nice. But do you have a result saying that $x_{k}$ will converge to the Gibbs distribution corresponding to $\theta_{\star}$ or something like that, e.g. ergodic mean etc.?

(5) Lemma 2 (Lemma 7) is very similar to the result for CSGLD (Theorem 1, Deng et al. (2020).) What's the technical novelty here?

---

> ### Author Response · Authors · 2021-11-19
> **Algorithm details**
>
> We appreciate the valuable comments.
>
> **Q1. Is the new target distribution required to be close to the original distribution?**
>
> No. Under the framework of dynamic importance sampling, we have
>
> $\text{E}[f(x)]=\int f(x) \underbrace{\pi(x)}_{\text{original density}}dx$
>
> $\qquad\quad=\int f(x)  \varpi_{\Psi_{\theta}}(x)  \underbrace{\frac{\pi(x)}{  \varpi_{\Psi_{\theta}}(x) }}_{\text{importance weight}}  d x,$
>
> where $\varpi_{\Psi_{\theta}}(x) $ is the modified density. The bias introduced by the accelerated escapes (from local traps) can be accounted for by the importance weight $\theta^{\zeta}$ as shown in the paper.
>
>
>
> $\newline$
>
> **Q2. Relation between $\Psi^{\zeta}_{\theta}(u)$  and $\widetilde U_{\Psi_{\theta}} (x)$.**
>
> Recall that $\Psi_{\theta}(U(x))= \sum_{i=1}^m \bigg(\theta(i-1)e^{(\log\theta(i)-\log\theta(i-1)) \frac{U(x)-u_{i-1}}{\Delta u}}\bigg) 1_{u_{i-1} < U(x) \leq u_i}$ and $\nabla_{x}\widetilde U_{\Psi_{\theta}}(x)= \left[1+ \frac{\zeta\tau}{\Delta u}  \left(\log \theta({J}(x))-\log\theta((J(x)-1)\vee 1) \right) \right] \nabla \widetilde U(x)$.
>
> By the chain rule, we have
>
> $\quad\nabla \log \Psi^{\zeta}_{\theta}(U(x))$.
>
> $=\zeta\frac{ d\log \Psi_{\theta}(U)}{d U} \nabla U(x)$
>
> $=\frac{\zeta}{\Delta u} (\log \theta(J(x))-\log\theta(J(x)-1)) \nabla U(x)$.
>
> This means that $\nabla U_{\Psi_{\theta}}(x)=\nabla U(x) + \tau \nabla \log \Psi^{\zeta}_{\theta}(U(x))$.
>
> $\nabla \widetilde U_{\Psi_{\theta}}(x)$ is a stochastic estimate of $\nabla U_{\Psi_{\theta}}(x)$ based on the stochastic energy $\widetilde U(x)$.
>
> $\newline$
>
> **Q3. How do you guarantee $\theta_k$ lives in a compact space?**
>
> Proving the compactness is straightforward and it is easy to show $\theta\in[0, 1]^m$. Recall that the stochastic approximation process follows that
>
> $$\theta(i) \leftarrow \theta(i)+\omega \theta(J_{\widetilde U}(x_{k+1})) \left(1_{i=J_{\widetilde U}(x_{k+1})}-{\theta}_{k}(i)\right)$$
>
> Informally, for any i, such that $\theta(i)>1$, negative penalties will be applied to ensure $\theta(i)\leq 1$; similarly, for any i, such that $\theta(i)<0$, positive penalties are always applied to ensure $\theta(i)\geq 0$.
>
> The **tricky part** is to show $\inf_{\Theta} \theta(i)>0$ instead of $\inf_{\Theta} \theta(i)\geq 0$ to ensure the stability in Eq.(29) is well-defined. We refer interested readers to the study of a recurrence property in [1] such that the particles return to a desired compact set often enough.
>
> [1] Convergence of the Wang-Landau algorithm. Math. Comput. 2015.
>
> $\newline$
>
> **Q4. How is the convergence of the weighted averaging estimators?**
>
> The convergence of the weighted averaging estimators can be shown in a similar fashion following Theorem 2 in [1].
>
> [1] A Contour Stochastic Gradient Langevin Dynamics Algorithm for Simulations of Multi-modal Distributions. NeurIPS'20.
>
>
> $\newline$
>
> **Q5. Difference between Lemma 2 and Theorem 1 in [1].**
>
> Recall that we are proving a stability condition $\langle h(\theta), \theta-\widehat \theta_{\star} \rangle \leq -\phi ||\theta-\widehat \theta_{\star}||^2,$
> where the equilibrium is $\widehat \theta_{\star}$, which follows that
>
> $\textcolor{blue}{\widehat \theta_{\star}= \theta_{\star} + \delta}$
>
> and $\delta$ is the maximum bias through the iterations with the order $O(\epsilon + \frac{1}{m} + \text{Var}(\xi(\theta)))$.
>
> Given the stability condition, we are able to show a direct result
>
> $\text{E}[||\theta_k-\widehat \theta_{\star}||^2]\leq O(\omega_k),$
>
> where the convergence analysis is much simplified and greatly **facilitates the proof of asymptotic normality**.
>
>
> By contrast, the stability condition in [1] follows that $\langle h(\theta), \theta- \theta_{\star} \rangle \leq -\phi ||\theta-\theta_{\star}||^2 + \delta.$
>
> The convergence theorem shows that
>
> $\text{E}[||\theta_k-\theta_{\star}||^2]\leq O(\omega_k) + \textcolor{red}{\frac{1}{\phi}}\delta.$
>
> Although both convergence results lead to the biases with the same order $O(\epsilon + \frac{1}{m} + \text{Var}(\xi(\theta)))$, the latter usually has a larger bias due to the **additional large coefficient $\frac{1}{\phi}$** as shown in the end of proof on page 18 [1]. A similar logic is used in [2] in proving the geometric ergodicity of gradient Langevin dynamics instead of Langevin diffusion [3] in improving the convergence rate of SGLD.
>
> [1] A Contour Stochastic Gradient Langevin Dynamics Algorithm for Simulations of Multi-modal Distributions. arXiv:2010.09800v1.
>
> [2] Global Convergence of Langevin Dynamics Based
> Algorithms for Nonconvex Optimization. NeurIPS'18
>
> [3] Non-Convex Learning via Stochastic Gradient Langevin Dynamics: A Nonasymptotic Analysis. COLT'17

---

> > ### Comment · Reviewer_9d9U · 2021-11-29
> > **response to authors response**
> >
> > I am satisfied with authors response. I have raised the score.

---

### Official Review · Reviewer_aTJ6 · 2021-11-02

**Correctness:** 2
**Technical Novelty And Significance:** 4
**Empirical Novelty And Significance:** 4
**Recommendation:** 6
**Confidence:** 4

**Main Review:**

The idea of Monte Carlo average of random-field functions from multiple chains in this paper is straightforward and easy to implement. The theoretical analysis seem to be solid too. However, I anticipate more discussion on the connection between empirical results and theoretical implications:
+ The analysis proves that ICSGLD improves estimation accuracy of parameter $\theta$ with non-infinitesimal step size. However, does parameter $\theta$ have any geometric interpretation (e.g. density of states)? Can we visualize the $\theta$ learned from ICSGLD and CSGLD? Can we empirically verify that the estimated variance is reduced?
+ Could the authors compare ICSGLD with different number of chains to further justify the theoretically result?
+ What are the main factors affecting the target $\theta_\ast$? Is it sensitive to model structure / model size / dataset ?
+ Is there any theoretical guide on the choice of hyper parameters e.g. $\zeta$? The first experiment uses $\zeta = 3e4$ and last experiment uses $\zeta = 3\times 10^6$. These values are significantly larger than other two experiments where $\zeta = 0.75$ and $\zeta = 20$.
How these parameters are selected in different experiments?
+ What is the recommended $\Delta u$ for neural network? What are the pros and cons for choosing small $\Delta u$?
+ What is the definition of preconditioned CSGLD and preconditioned ICSGLD? In section 5.3, authors cite [Deng et al., 2020b] for preconditioned CSGLD. However, it seems this paper doesn't contain preconditioned version of CSGLD.

Others:
+ How KL divergence is estimated in Figure 6?
+ It will be appreciated if the code is provided for reproduction.


**Summary Of The Paper:**

This paper samples from distributions with complex energy landscapes by the powerful contour stochastic gradient Langevin dynamics (CSGLD) methods. In order to reduce the variance of random-field function, this paper proposes a simple but effective method, that combines  random-field functions from multiple chains. Since the parameter $\theta$ typically has small size, the communication overhead is marginal. This paper show that compared to CSGLD which only has one chain, the variance ICSGLD with P chains and same computation budget is asymptotically more efficient. The authors evaluate ICSGLD with multiple experiments.

**Summary Of The Review:**

This paper proposes an interesting extension to CSGLD whose advantages are shown both theoretically and empirically. However, I believe many questions deserve further investigation before applying ICSGLD into practical machine learning. I would like to hear authors' response on how theoretical implications are connected to practical concerns.

---

> ### Comment · Reviewer_aTJ6 · 2021-11-12
> **Additional Questions**
>
> The authors claim that `ζ can be tuned to generate proper bouncy moves` to avoid local-trap problems, and authors uses heavily tuned parameter $\zeta$ in different experiments.
> However, I believe the above quoted claim is not correct, at least in noiseless case.
>
> At page 16, authors shows following formula
>
> $\mathbf{x}\_{k+1}=\mathbf{x}\_{k}-\epsilon\_{k+1} \frac{N}{n}\left[1+\zeta \tau \frac{\log \theta\_{k}\left(\tilde{J}\left(\mathbf{x}\_{k}\right)\right)-\log \theta\_{k}\left(\left(\tilde{J}\left(\mathbf{x}\_{k}\right)-1\right) \vee 1\right)}{\Delta u}\right] \nabla\_{\mathbf{x}} \widetilde{U}\left(\mathbf{x}\_{k}\right)+\sqrt{2 \tau \epsilon\_{k+1}} w\_{k+1},$
>
> which give rises to the gradient multiplier
> $1+\zeta \tau \frac{\log \theta\_{k}\left(\tilde{J}\left(\mathbf{x}\_{k}\right)\right)-\log \theta\_{k}\left(\left(\tilde{J}\left(\mathbf{x}\_{k}\right)-1\right) \vee 1\right)}{\Delta u}$.
>
> At first glance, $\zeta$ does directly control the scalar gradient multiplier, therefore small value of ζ would lead to `gradient multiplier uniformly “equals” to 1` and `self-adjusting mechanism works only if we choose a large enough ζ` as authors argued in B.1.3.
>
> However, it turned out $\theta_k$ are also controlled by $\zeta$ in the way that the target value is $\theta\_{\star}(i) \propto \boldsymbol{\theta}\_{\infty}^{\frac{1}{\zeta}}(i)$. If we introduce $\zeta$ dependence of $\theta_k$ in gradient multiplier, we can see that two effects cancel with each other, and $\zeta$ has no directly impact on gradient multiplier.
>
> More rigorously, consider the proposal distribution $\varpi\_{\Psi\_{\boldsymbol{\theta}}}(\mathbf{x}) \propto \pi(\mathbf{x}) / \Psi\_{\boldsymbol{\theta}}^{\zeta}(U(\mathbf{x}))$ as defined in eq. (2), at perfect estimation case, i.e. $\theta=\theta\_{\star}(i) \propto \boldsymbol{\theta}\_{\infty}^{\frac{1}{\zeta}}(i)$, we have that the proposal distribution is independent of $\zeta$.
>
> The above observation negates authors argument and gives rise to a question regarding the true effect of $\zeta$.
>
> I kindly encourage authors to respond to above comments and explain how different $\zeta$ affect the evaluation. Additional ablation study with different $\zeta$ would be appreciated.

---

> ### Comment · Reviewer_aTJ6 · 2021-11-17
> **Update**
>
> The true impact of $\zeta$ is still unclear due to lack of author response.
> I feel this problem notably hurts the significance and clarity of this paper, therefore I updated my score accordingly.

---

> > ### Author Response · Authors · 2021-11-18
> > **$\zeta$ facilitates the self-adjusting mechanism and affects the convergence rate**
> >
> > Sorry for the late reply. We appreciate the reviewer for raising the insightful question.
> >
> > First, we would like to point out that $\zeta$ will not affect the algorithm (gradient multiplier) when $\theta$ has reached its equilibrium, which agrees with the opinion of the reviewer. However, the **convergence speed/rate will be affected** based on different $\zeta$. Next, we detail the reasons from two perspectives.
> >
> >
> > Recall that the adaptive sampling follows that
> >
> > \begin{equation*}
> >   x\leftarrow x - \epsilon \frac{N}{n}  \left[1+
> >    \zeta\tau  \underbrace{\frac{\log {\theta}(J(x))-\log {\theta}(J(x)-1)}{\Delta u}}_{\text{sub-multiplier } \mathcal{S}} \right] \nabla \widetilde U(x)+\sqrt{2 \tau \epsilon} w,
> > \end{equation*}
> >
> >
> > **Argument 1: How $\zeta$ facilitates the self-adjusting mechanism**
> >
> >
> > By the algorithmic nature [1], the local trap issue has been largely alleviated given the ideal equilibrium $\theta_{\star}$. However, **$\theta_{\star}$ is not known a priori**. When **$\theta$ deviates from $\theta_{\star}$** and the local trap issue is severe, the particles easily get stuck in local optima or saddle points.
> >
> > ***Local optima*** When a particle gets stuck in a local optimum for a long time, a low energy region usually has a larger density than high energy regions, hence we have a negative sub-multiplier $\mathcal{S}<0$. Given a large enough $\zeta$, the multiplier $1+\zeta\tau \mathcal{S}$ is sufficiently small or negative and a particle potentially bounces back to high energy regions and accelerates the convergence. With a small $\zeta$, however, it may take more time to escape local traps and limits the self-adjusting mechanism.
> >
> > ***Saddle points*** The particle behaves quite in the opposite and a larger $\zeta$ enables a larger learning rate to traverse saddle points.
> >
> >
> > **Argument 2: How $\zeta$ yields faster convergence rate of $\theta$ and sampling**
> > From another perspective, as shown in Lemma 1, $\theta_{\star} \propto \theta_{\infty}^{\frac{1}{\zeta}}$ and a larger value of $\zeta$ leads to a **smoother $\theta_{\star}$, which is easier to approximate**. Theoretically, as discussed in Eq.(29) on page 20, the mean field equation follows that
> > \begin{equation}
> >     h_i(\theta) \approx \widehat Z_{\zeta, \cdot} \left[\theta_{\star}(i) -\theta(i)\right]
> > \end{equation}
> >
> > where different $\zeta$ yields a different value for $\widehat Z^{\cdot}_{\zeta, \cdot}$ and hence **a different convergence (contraction) rate** for stochastic approximation, which also affects the convergence of sampling algorithms. In other words, we should choose $\zeta$ that yields large enough contraction rate as long as it doesn't cause stability issues.
> >
> > In summary, a fine-tuned $\zeta$ facilitates the self-adjusting mechanism and improves the convergence rate of stochastic approximation to the equilibrium $\theta_{\star}$, where the density of states $\theta_{\star}^{\zeta}$ possesses the ideal properties to help particles escape local traps [1] and accelerates the convergence of sampling.
> >
> > We thank the reviewer again for your interest and time investment. Hope that we have addressed your concerns. We would like to respond to any further concerns that you may have to improve the manuscript.
> >
> > [1] Multiple-range Random Walk Algorithm to Calculate the Density of States. Phy. Rev. Let., 2001.

---

> > > ### Comment · Reviewer_aTJ6 · 2021-11-24
> > > **Discussion on ζ**
> > >
> > > > the convergence speed/rate will be affected based on different $\zeta$
> > >
> > > I fully agree. However, I feel the detail of this dependence is not well understood.
> > > What's make things worse is that authors use lots of non rigorous arguments.
> > >
> > > > Given a large enough $\zeta$, the multiplier $1+\zeta \tau S$ is sufficiently small ... With a small $\zeta$, however, it may take more time to escape local traps and limits the self-adjusting mechanism.
> > >
> > > In order to make sense out of above sentence, we have to implicitly assume  $1+\zeta \tau S$ to be monotonically decrease as a function of $\zeta$. However, that might not always be true. S depends on $\theta$ whose dynamics and optimal value are controlled by $\zeta$ in a non trivial way.
> > >
> > > > The particle behaves quite in the opposite and a larger $\zeta$ enables a larger learning rate to traverse saddle points.
> > >
> > > Again, we have to assume $1+\zeta \tau S$ monotonically increase as a function of $\zeta$.
> > >
> > > > a larger value of $\zeta$, leads to a smoother $\theta_\ast$, which is easier to approximate
> > >
> > > This is partial and misleading.
> > > If all we want is smooth $\theta_\ast$, then why we don't use $\zeta=\infty$ such that $\theta_\ast$ is uniform and don't have to be estimated at all?
> > >
> > > The authors ignore that when $\zeta$ is large, error in $\theta$ will be enlarged and be reflected in error of proposal distribution $\varpi\_{\Psi\_{\boldsymbol{\theta}}}(\mathbf{x}) \propto \pi(\mathbf{x}) / \Psi\_{\boldsymbol{\theta}}^{\zeta}(U(\mathbf{x}))$.
> > >
> > > > Theoretically, as discussed in Eq.(29) on page 20... different $\zeta$ yields a different value for $\hat{Z}\_\zeta$ and hence a different convergence (contraction) rate
> > >
> > > True. But the definition of $\hat{Z}\_\zeta$ doesn't has any obvious monotone structure.
> > > I also didn't see any qualitative analysis on how $\hat{Z}\_\zeta$ depend on $\zeta$.
> > >
> > > > we should choose $\zeta$ that yields large enough contraction rate as long as it doesn't cause stability issues
> > >
> > > False. Due to the noise, $\theta$ will not converge with finite step size, no matter how large the contraction rate is. Tuning $\zeta$ would affect both contraction rate and noise magnitude. Maximizing contraction rate doesn't necessarily accelerate convergence.
> > >
> > > In summary, the effect of $\zeta$ remains a mystery to me.
> > > Although I agree that $\zeta$ has some impact on the algorithm, I am not convinced that large $\zeta$ would accelerate the convergence.

---

> ### Author Response · Authors · 2021-11-18
> **Hyperparameter tuning, geometric interpretation, and code reproduction**
>
> Q1. Hyperparameter tuning of $\zeta$.
>
> A fine-tuned $\zeta$ usually leads to a small or even slightly negative learning rate in local optima to avoid local-trap problems. Our empirical experience shows that tuning $\zeta$ to limit the lowest gradient multiplier $1+\tau\zeta \mathcal{S}$ around $-1$ achieves remarkable performance. Further increasing $\zeta$ may cause stability issues of the algorithm and over-visit the tails of the distributions.
>
> Theoretically, a good choice of $\zeta$ does maximize the contraction rate for the convergence of the underlying mean-field equation and thus accelerates the simulation. We will include more discussions in section B.1.1 in the next revision.
>
> $\newline$
>
> Q2. Geometric interpretation (e.g. density of states)
>
> Yes. The algorithm is inspired by the Wang-Landau algorithm [1], which proposes to estimate the density of states (or spectral density) via SGLD samples and generates a flat histogram in the energy space. Note that the Wang-Landau algorithm is simulated based on the gradient-free samplers and requires more smoothness on the algorithm design to adapt to gradient-based samplers.
>
> [1] Multiple-range Random Walk Algorithm to Calculate the Density of States. Phy. Rev. Let., 2001.
>
>
>
>
>
> $\newline$
>
> Q3. Hyperparameter tuning of $\Delta u$.
>
> Setting a large $\Delta u$ is easier to estimate, which, however, may sacrifice the self-adjusting mechanism and lead to a larger gap (error) between the equilibrium of stochastic approximation and the ideal $\theta_{\infty}^{\frac{1}{\zeta}}$, as illustrated in Lemmas 1 and 2. On the contrary, setting a small $\Delta u$ may lead to a large variance and require more effort to estimate. In practice, we find a good bias-variance trade-off can be obtained with around 100-10000 partitions.
>
>
> $\newline$
>
>
> Q4. No preconditioned version of CSGLD.
>
> Thanks for pointing out this issue. We will clarify it in the next version, where the preconditioned CSGLD proposes a preconditioner following [1] to CSGLD.
>
> [1] Preconditioned Stochastic Gradient Langevin Dynamics for Deep Neural Networks. AAAI'16.
>
> $\newline$
>
> Q5. Code for reproduction.
>
> To help reproduce all the important results and benefit the community for MCMC computations, we have made all the code available in the supplementary file. Sorry that it does take some time to clean our code, which delays our response.
>
> $\newline$
>
> Q6. How is KL divergence estimated?
>
> It is implemented based on [1]. The code is provided in the file **simulations/plot/Plot.ipynb** in the supplementary file.
>
> [1] Pérez-Cruz, F. Kullback-Leibler divergence estimation of continuous distributions IEEE International Symposium on Information Theory, 2008.

---

> > ### Comment · Reviewer_aTJ6 · 2021-11-24
> > **Property of θ**
> >
> > I thank authors for their response.
> >
> > Although the authors explain the geometric interpretation perfectly, they seem to miss my following questions.
> >
> > > Can we visualize the $\theta$ learned from ICSGLD and CSGLD?
> >
> > > Can we empirically verify that the estimated variance is reduced?
> >
> > > What are the main factors affecting the target $\theta_\ast$? Is it sensitive to model structure / model size / dataset ?
> >
> > I want to add a few new questions about the properties of $\theta$.
> >
> > 1. How similar are the state densities of two different neural networks?
> > 2. If the $\theta$ deviations of different neural networks are not large, can we use the pre-trained $\theta$ to avoid CSGLD during training?
> > 3. If the $\theta$ deviations of different neural networks are large, how do different hyperparameters affect this value?

---

> > > ### Author Response · Authors · 2021-11-29
> > > **Visualization of $\theta$ and variance reduction**
> > >
> > > **Q1. Can we visualize $\theta$ learned from CSGLD and ICSGLD and empirically verify that the estimated variance is reduced?**
> > >
> > >
> > > To compare the $\theta$'s learned from CSGLD and ICSGLD, we try to simulate from a Gaussian mixture distribution $0.4 N(-6, 1) + 0.6 N(4, 1)$. We fix $\zeta=0.9$ and $\Delta u=1$. We run ICSGLD with 1,000,000 iterations based on 10 interacting parallel chains and run CSGLD with 10,000,000 iterations using a single chain. The rest of the settings follows from the experimental setup in section 4.1 [1].
> > >
> > > We present our results at https://pasteboard.co/bBnIvsAbLhRg.png, where the left plot indicates that both estimates of $\theta^{\zeta}$ (by CSGLD and ICSGLD) converge to the ground truth of the density of states.
> > >
> > > To measure the variance of the estimates, we repeated the experiments 9 times and reported the root mean squared error (RMSE) of  $\theta^{\zeta}$. The right plot (at https://pasteboard.co/bBnIvsAbLhRg.png) indicates that ICSGLD yields a much smaller variance than CSGLD, but with the same computational budget. This shows the clear advantage of ICSGLD (many interacting short runs) over CSGLD (a single long run).
> > >
> > > The implementation details are provided in https://pasteboard.co/n52FZUJFfk3u.png.
> > >
> > > [1] A Contour Stochastic Gradient Langevin Dynamics Algorithm for Simulations of Multi-modal Distributions. NeurIPS'20.
> > >
> > > $\newline$
> > >
> > > **Q2. Main factors affecting the target $\theta_{\star}$.**
> > >
> > > Recall Eq.(9), we have $\theta_{\star}(\cdot)\propto\theta_{\infty}^{\frac{1}{\zeta}}(\cdot)$. Therefore, $\theta_{\star}$ depends on both $\zeta$ and $\theta_{\infty}$. Note that $\theta_{\infty}$  characterizes the density of states or spectral density. The neural network structure (widths, depths) will affect the energy landscape and thus $\theta_{\infty}$. Moreover, more data points often lead to a more concentrated posterior, which increases the density at low energy regions.
> > >
> > > $\newline$
> > >
> > > **Q3. Additional properties of $\theta_{\infty}$.**
> > >
> > > **Q3.1 How similar are the state densities of two different neural networks?**
> > >
> > > If both networks can approximate the same dataset (or the underlying function) equally well, their densities of states should be much similar (by assuming that the non-informative prior $\pi(x) \propto 1$ is used, where $x$ refers to the weights of the neural network).
> > >
> > >
> > > **Q3.2 If the deviations of different neural networks are not large, can we use the pre-trained $\theta$ to avoid CSGLD during training?**
> > >
> > > Thanks for your thoughtful question. Yes, the pre-trained $\theta$ from one network can be used for another network. I believe this feature will be extremely useful for transfer learning, and we will try it in the future.
> > >
> > > **Q3.3 If the deviations of different neural networks are large, how do different hyperparameters affect this value?**
> > >
> > >
> > > As mentioned previously, if two networks have about the same approximation capability, they should have similar estimates of $\theta$ and the same hyperparameters can be used for them.
> > >
> > > Also, we note that $\zeta$ is an enhancing parameter for sample space exploration, but an excessively large value of $\zeta$ might adversely affect the convergence of the algorithm. Therefore, for a new problem, we can start with a large value of $\zeta$ for sample space exploration, and then refine it later towards fast convergence.
> > >
> > >
> > > We thank the reviewer again for the fruitful comments on our work and we hope that we have answered all the concerns.

---

> > > > ### Comment · Reviewer_aTJ6 · 2021-12-01
> > > > **Response**
> > > >
> > > > I thank the authors for their response.
> > > >
> > > > Although the additional experiment was only in the toy model, it did answer my question.
> > > > Although how $\zeta$ should be tuned is still a mystery, I agree that it is an additional hyperparameter that affects convergence.
> > > > Although the property of $\theta_\ast$ is not fully understood, I think this is not directly linked with main contribution of this paper, and should be left for future work.
> > > > Overall, I believe the method proposed in the paper is simple, effective and worth of publication.
> > > > Therefore, I set my final score to 6.

---

### Official Review · Reviewer_mVcL · 2021-11-03

**Correctness:** 4
**Technical Novelty And Significance:** 4
**Empirical Novelty And Significance:** 4
**Recommendation:** 6
**Confidence:** 3

**Main Review:**

I liked reading the paper. Overall, the work addresses a relevant problem and provides a nice introduction to previously existing solutions.

In my opinion, the contributions are significant and the proposed algorithm is presented in a clear way. The analysis is sound and backed up by numerical evidence in different settings. I believe that the paper is worth publication.

However, I have some minor questions that require a clarification from the authors.

(i) I do not understand how the parameter $\zeta$ is tuned in practice and I would like to have more details on this procedure in addition to the comments provided in appendix B.1.3. This point seems to play an important role, e.g., in escaping from local regions of the landscape in Figure 2, while Appendix D.1 only says that $\zeta$ is fixed to $3e4$. Since in the other cases (appendix D.2, D.3) $\eta$ is fixed to much smaller values (namely, $0.75$ and $20$) I would like to have more intuition on this.

(ii) I would like to have some clarifications on the link between the second footnote on page 7 and the sentence it refers to.

**Summary Of The Paper:**

This paper addresses the problem of sampling from distributions with complex energy landscapes. The authors propose an extension of the contour Stochastic Gradient Langevin dynamics (CSGLD) sampler to efficiently simulate from big-data distributions. The extension is called “interacting” since $P$ different CSGLD are updated simultaneously and the random-field function is obtained as a Monte-Carlo average over the random field functions of these multiple chains run in parallel. This procedure allows to reduce the variance of the self-adapting parameter $\theta$ improving the marginal energy likelihood estimate, and is well-suited for distributed computing. The different chains only share the low-dimensional vector $\theta$ and do not share the model parameters, that are typically high dimensional. The authors prove that the proposed algorithm is asymptotically more efficient than its single-chain counterpart with the same computational budget. This theorem holds for a step size that decreases polynomially in time. Finally, they compare their algorithm to alternative methods on different tasks (e.g., mode exploration on MNIST dataset, uncertainty estimation) achieving competitive performances.

**Summary Of The Review:**

I believe that the paper is worth of publication since it presents an efficient importance sampling method that scales to big data problems and possibly leads to additional future investigations.

---

> ### Author Response · Authors · 2021-11-19
> **Hyperparameter tuning of $\zeta$**
>
>
> We appreciate the valuable comments.
>
> **Q1. Clarification on the link between the second footnote and the sentence it refers to.**
>
> It is known that setting a low temperature easily leads to the local trap issue. However, setting $\tau=0.1$ in the MNIST example is not meant to intentionally create challenges for the samplers. The cold posterior effect is quite standard in training deep learning models due to the use of data augmentation techniques [1,2].
>
>
> **Q2. Hyperparameter tuning of $\zeta$.**
>
> A fine-tuned $\zeta$ usually leads to a small enough or even slightly negative learning rate in local optima to avoid local-trap problems. Our empirical experience shows that tuning $\zeta$ to limit the lowest gradient multiplier $1+\tau\zeta \mathcal{S}$ around $-1$ achieves remarkable performance, where $\mathcal{S}$ is the sub-multiplier defined in our response to *reviewer aTJ6*. Further increasing $\zeta$ may cause stability issues of the algorithm and over-visit the tails of the distributions.
>
> Theoretically, a good choice of $\zeta$ does maximize the contraction rate for the convergence of the underlying mean-field equation in Eq.(29) and thus accelerates the simulation. We will include more discussions in section B.1.1 in the next revision.
>
> **Q3. Why a large $\zeta$ is often used in deep learning models**
>
> It can be explained from the perspectives of smoothness of equilibrium and trajectory behavior:
>
> **Smoothness of equilibrium**: Due to a large number of data points and the popular data augmentation techniques in deep learning tasks, the posterior distribution yields the cold posterior effect [1,2] and is extremely concentrated around the global optima with an excessively large Lipschitz constant. Recall that the density of states follow that $\theta_{\infty}(i)=\int_{X_i} \pi(x)dx$ for $i\in\\{1,2,\cdots, m\\}$, which is extensively concentrated around a few partitions and is rather unbalanced. Through a large $\zeta$, the equilibrium $\theta_{\star}$ follows that $\theta_{\star}\propto \theta_{\infty}^{\frac{1}{\zeta}}$, where $\theta_{\infty}$ acts as the density of states, hence a larger $\zeta$ leads to a smoother $\theta_{\star}$, which is much easier to estimate.
>
> **Trajectory Behavior**: When a particle gets stuck in a local optimum, a low energy region usually has a larger density than high energy regions, hence we have a negative sub-multiplier $\mathcal{S}$ as defined in our response to *reviewer aTJ6*. Given a large enough $\zeta$, the multiplier $1+\zeta\tau \mathcal{S}$ is **sufficiently small or negative and a particle potentially bounces back to high energy regions** and thus accelerates the convergence. With a small $\zeta$, however, it may take more time to escape local traps and limit the self-adjusting mechanism.
>
> In summary, as pointed out by *reviewer aTJ6*, although $\zeta$ does not affect the algorithm when $\theta=\theta_{\star}$, a fine-tuned $\zeta$ still accelerates the convergence rate of the algorithm and facilitates the particle to escape from local traps.
>
> [1] A Statistical Theory of Cold Posteriors in Deep Neural Networks. ICLR, 2021.
>
> [2] How Good is the Bayes Posterior in Deep Neural Networks Really? ICML, 2020.

---

> > ### Comment · Reviewer_mVcL · 2021-11-30
> > **Response to authors**
> >
> > I thank the authors for their clarifications. Overall, the paper is well written and the contribution is sufficient for publication. Therefore, I am keeping my score to 6.

---

### Official Review · Reviewer_CBb1 · 2021-11-03

**Correctness:** 4
**Technical Novelty And Significance:** 3
**Empirical Novelty And Significance:** 3
**Recommendation:** 8
**Confidence:** 3

**Main Review:**

My main concern is the clarity of the paper. At some points, the paper looks like a riddle where you are looking for some notation and definitions. Some points of the exposition lack intuition.
- The choice of the piecewise continuous approximation of the target density and the function H is not apparent at all. I would suggest sharing a piece of intuition with the reader to motivate the usage of these functions early on. The clarity is also hardened by the fact that the original paper by [Deng, 2020] also doesn't provide any intuition for the choice.
- The proposed algorithm is presented in the background section together with the algorithm by [Deng, 2020]. It would be much easier for the reader if the authors would highlight the difference between the original algorithm and its modification. For instance, they could say that the main difference between algorithms is in the estimation of parameters $\theta$ and using all of the samples instead of a single chain or even highlight it explicitly in the formulas by using a different color.
- Pictures in Figure 1 are not informative. The pictures depict different aspects of interacting chains algorithms. Fig. 1a represents the jumps in the state space, and Fig. 1b represents the mutual update of parameters. The authors use Fig. 1b as a motivation for better mixing properties, claiming that unlike the replica-exchange method their algorithm shares "messages" between samples at every step. However, one can argue that the replica-exchange method also shares information between samples at every step by running the accept/reject test trying to swap the samples.

Another major issue is that the authors do not compare their algorithm against CSGLD (algorithm by [Deng, 2020]) in one of the main experiments: Bayesian networks on CIFAR-100 and SVHN. Due to the close relations between algorithms, such kind of comparison is essential for the performed empirical study.


**Summary Of The Paper:**

The paper proposes an MCMC algorithm containing two key ideas: the approximation of the target density with a simpler function (proposed by [Deng, 2020]) and the parallel simulation of many chains that allow for better capturing the global properties of the target density. To be more precise, the main contribution of the paper is the extension of the algorithm by [Deng, 2020] to the case of multiple chains.

The proposed extension goes as follows. The original algorithm by [Deng, 2020] approximates the target density with piecewise continuous functions using the level sets (slicing the values of the energy function). These functions are parameterized by vector $\theta$, which needs to be defined. In practice, $\theta$ is estimated using the samples from the chain. The current paper then proposes to run many chains in parallel and use all of the available samples for the estimation.

The efficiency of the proposed technique is analyzed both theoretically and empirically. For theoretical analysis, the authors provide convergence speed and asymptotic distribution of the residuals $\theta - \theta^*$. Overall, the analysis favors the proposed method over the original algorithm. The empirical comparison is made on several tasks. Except for the toy task, the authors consider learning policy for contextual bandit problem and sampling from the posterior distribution of a neural network.

[Deng, 2020]: Deng, Wei, Guang Lin, and Faming Liang. "A Contour Stochastic Gradient Langevin Dynamics Algorithm for Simulations of Multi-modal Distributions." arXiv preprint arXiv:2010.09800 (2020).

**Summary Of The Review:**

This submission is a follow-up to [Deng, 2020] that extends the previous work to the case of multiple parallel chains. Although the extension seems quite straightforward, the authors provide a theoretical analysis of the proposed scheme and perform its empirical study. The main issues are the clarity of the exposition and one inconsistency in the experimental section.

[Deng, 2020]: Deng, Wei, Guang Lin, and Faming Liang. "A Contour Stochastic Gradient Langevin Dynamics Algorithm for Simulations of Multi-modal Distributions." arXiv preprint arXiv:2010.09800 (2020).

---

> ### Author Response · Authors · 2021-11-19
> **Intuition and clarification of the modification**
>
>
> We appreciate the valuable comments.
>
> **Q1. The motivation of the piece-wise continuous approximation and the modified random field function.**
>
> **Piece-wise continuous approximation:** The algorithm is motivated by the Wang-Landau algorithm [1], which is a flat histogram algorithm and proposes to penalize over-revisited regions with a large weight or **bouncy moves**. The ideal weight is the $\theta_{\star}^{\zeta}$, which closely relates to the density of states (or spectral density) in statistical physics.
>
> The original algorithm is based on the **gradient-free** Metropolis sampler with a **piece-wise constant** approximation, which, however, doesn't adapt to the gradient-based Langevin sampler (taking gradient to a constant is 0). As such, we propose the **piece-wise continuous** approximation to resolve that issue.
>
> **Intuition of the random field function $H$:** The stochastic approximation of the latent variable $\theta_k$ follows that
> \begin{equation}
> \begin{split}
>     \theta_{k+1}&=\theta_k + \omega_{k+1} H(\theta_k, x_{k+1})\\
>                 &=\theta_k + \omega_{k+1} h(\theta_k) + \text{perturbation} + \text{martingale noise}\\
> \end{split}
> \end{equation}
>
> Setting $\omega_{k}\rightarrow 0$ and eliminating the perturbation and martingale noise, the ideal case is to study an ordinary differential equation (ODE) as follows **$\partial \theta=h(\theta)$**.
>
> The equilibrium $\theta_{\star}$ of the ODE satisfies that $h(\theta_{\star})=0$. Following Eq.(22) in the appendix, we can obtain the desired $\theta_{\star}$ based on the random field function $H(\theta, x)$.
>
> [1] Multiple-range Random Walk Algorithm to Calculate the Density of States. Physical Review Letters, 2001.
>
> $\newline$
>
>
> **Q2. Highlight the difference between the original algorithm and its modification**
>
> The algorithm includes two contributions compared to the original CSGLD algorithm.
>
> The **major novelty** is the interacting random field function $\widetilde H(\theta, x^{\bigotimes P})=\frac{1}{P}\sum_{p=1}^P \widetilde H(\theta, x^{(p)})$. Note that the martingale noise is a crucial item to affect the convergence of stochastic approximation and the **conditional independence** of each parallel chain ensures an **ideal variance reduction** of the interacting random field function.
>
> The **second novelty is the scalability** in big data problems, where the following is a partial restatement of section B.1.3.
>
> The key to the success of (I)CSGLD is to generate sufficiently strong **bouncy moves** to escape local traps.  Take the CIFAR100 experiments for example and set the fine-tuned $\zeta=$3e6 as in [1].
>
> The original stochastic approximation (SA) in [1] follows that
>
> $${\theta}(i)\leftarrow {\theta}(i)+\omega \textcolor{red}{{\theta}^{\zeta}}(J(x))\left(1_{i=J(x)}-{\theta}(i)\right).$$
>
>
> Since $\theta(i)<1$ for any $i\in\\{1,\cdots, m\\}$, $\theta(i)^{\zeta}$ **is essentially 0** for such a large $\zeta=3e6$, which means that **the original SA fails to optimize when $\zeta$ is large**, which **limits the update of bouncy moves**.
>
> Our newly proposed SA scheme
>
> $${\theta}(i)\leftarrow {\theta}(i)+\omega \textcolor{blue}{\theta}(J(x))\left(1_{i=J(x)}-{\theta}(i)\right).$$
>
> is more independent of $\zeta$ and proposes to converge to a much **smoother equilibrium** $\theta_{\infty}^{1/\zeta}$ instead of $\theta_{\infty}$, where $\theta_{\infty}$ is the energy PDF or density of states. It is clear that $\theta_{\infty}^{1/\zeta}$ is easier to estimate than $\theta_{\infty}$ for a large $\zeta$.
>
> [1] A Contour Stochastic Gradient Langevin Dynamics Algorithm for Simulations of Multi-modal Distributions. NeurIPS'20.
>
> $\newline$
>
> **Q3. Figure 1 is not informative.**
>
> Conducting accept/reject tests alone cannot accelerate parallel tempering in non-convex scenarios. The success of parallel tempering (PT) hinges on efficient swapping schemes to balance exploitation and exploration.  Standard schemes that attempt to swap neighboring chains yield a round trip time $O(P^2)$, where the round trip is the time used for a particle to move from chain 1 to chain P and then back to chain 1. The state-of-art scheme [1] yields an appealing round trip time of $O(P)$ (**with a large constant though**) given sufficient many chains with high-enough acceptance rates. However, given finite $P$ chains with the low bias-corrected acceptance rates in big data, the **quadratic round trip time $O(P^2)$** is still unavoidable.
>
> By contrast, our proposed algorithm only requires to average low-dimensional latent vectors every iteration with an ideal variance reduction. Because communication of **model parameters in PT-SGLD** is much more costly than that of **latent vectors in ICSGLD** and PT-SGLD requires a long round trip time $O(P^2)$ to balance exploitation and exploration. Figure 1 naturally implies a high efficiency of ICSGLD.
>
>
> [1] Non-Reversible Parallel Tempering: a Scalable Highly Parallel MCMC Scheme. arXiv:1905.02939v4.

---

> > ### Author Response · Authors · 2021-11-22
> > **ICSGHMC$\times$P4 vs CSGHMC$\times$P4**
> >
> > As suggested by the reviewer for a fair comparison, we have implemented CSGHMC$\times$P4 on CIFAR100 dataset, which is a naively parallel implementation of CSGHMC without interacting latent variables. Due to the time limit, we were only able to run results based on ResNet20 and 32 models and we will include the rest of them in the next revision.
> >
> > ------------------------------------------------------------------------
> >                           ACC(%)     NLL    Brier(%)
> > ------------------------------------------------------------------------
> > ResNet20  CSGHMC$\times$P4  $\quad$76.29$\pm$0.19$\quad$8088$\pm$49$\quad$2.51$\pm$0.10
> >
> > ResNet20  ICSGHMC$\times$P4$\quad$76.39$\pm$0.15$\quad$8076$\pm$31$\quad$2.52$\pm$0.07
> >
> >
> >
> >
> > ------------------------------------------------------------------------
> >                            ACC(%)     NLL    Brier(%)
> > ------------------------------------------------------------------------
> > ResNet32  CSGHMC$\times$P4  $\quad$78.69$\pm$0.23$\quad$7379$\pm$38$\quad$2.92$\pm$0.19
> >
> > ResNet32  ICSGHMC$\times$P4$\quad$78.79$\pm$0.16$\quad$7384$\pm$29$\quad$2.73$\pm$0.12
> >
> >
> > Since ResNet models are highly optimized, only a slight improvement can be obtained and ICSGHMC$\times$P4 cannot obtain significantly better results than CSGHMC$\times$P4. Nevertheless, we observe in the experiments that CSGHMC$\times$P4 **may become unstable during the training and causes mediocre results**, a typical drawback ($\textcolor{red}{\text{large variance}}$) of importance sampling in practice. By contrast, ICSGHMC$\times$P4 consistently performs remarkably due to the ideal variance reduction through interacting latent variables based on conditional independent samples.

---

> > > ### Comment · Reviewer_CBb1 · 2021-11-29
> > > **Response to the authors**
> > >
> > > Thank you for the detailed comments.
> > > I hope the authors can make the corresponding changes to the work and extend the current comparison to the previous work. I would suggest demonstrating the instabilities of CSGHMC mentioned above quantitatively rather than simply describing them in the text.
> > >
> > > In general, I'm leaning toward acceptance and would like to increase my score.

---

> > > > ### Author Response · Authors · 2021-11-30
> > > > **Response**
> > > >
> > > > Thanks for your suggestions. We will make the corresponding changes and include the quantitative studies on the instabilities of CSGHMC in the next revision.

---

### Decision · Program_Chairs · 2022-01-20

**Decision:**

Accept (Poster)

**Comment:**

This paper proposes a new variant of a stochastic gradient Langevin dynamics sampler that relies on two key ideas: approximation of the target density with a simpler function (as in [Deng, 2020]) and the parallel simulation of many chains. The authors also prove that their approach can be theoretically more efficient than a single-chain algorithm.

The reviewers see the contribution as significant although they did raise some concerns regarding the clarity of the paper. Since these concerns do not appear to be major, I recommend acceptance but I advise the authors to address the comments of the reviewers to maximize the impact of the paper.